# The ClpX protease is essential for inactivating the CI master repressor and completing prophage induction in *Staphylococcus aureus*

Mohammed A. Thabet [1,2], José R. Penadés [3] & Andreas F. Haag [1,4] ✉

Bacteriophages (phages) are the most abundant biological entities on Earth, exerting a significant influence on the dissemination of bacterial virulence, pathogenicity, and antimicrobial resistance. Temperate phages integrate into the bacterial chromosome in a dormant state through intricate regulatory mechanisms. These mechanisms repress lytic genes while facilitating the expression of integrase and the CI master repressor. Upon bacterial SOS response activation, the CI repressor undergoes auto-cleavage, producing two fragments with the N-terminal domain (NTD) retaining significant DNA-binding ability. The process of relieving CI NTD repression, essential for prophage induction, remains unknown. Here we show a specific interaction between the ClpX protease and CI NTD repressor fragment of phages Φ11 and 80α in *Staphylococcus aureus*. This interaction is necessary and sufficient for prophage activation after SOS-mediated CI auto-cleavage, defining the final stage in the prophage induction cascade. Our findings unveil unexpected roles of bacterial protease ClpX in phage biology.

Bacteriophages, or phages, are the most abundant biological entities on Earth[1] and play a crucial role in horizontal gene transfer, including the dissemination of antimicrobial resistance genes and virulence factors[2]. Temperate phages, such as the model phage λ establish a complex relationship with their bacterial host, where they can either replicate and kill their host cell (lytic cycle) or integrate into their host's genome and be passed on vertically (lysogenic cycle) until they become reactivated upon defined environmental cues. Lysogeny is controlled by a phage repressor called CI in λ, which represses the transcription of genes required for phage excision, replication, packaging, and host cell lysis as well as its own transcription[3,4]. Upon induction, the prophage produces the lytic regulator protein Cro, which stabilises the lytic cycle by blocking transcription from the $P_{RM}$ promoter driving *cI* transcription[3]. The expression levels of CI and Cro

are finely balanced to ensure tight repression of the phage lytic cycle while enabling rapid activation upon sensing the appropriate stimulus.

The bacterial SOS response is a key signal that activates the phage lytic cycle. It is triggered by DNA damage and coordinates the expression of genes involved in DNA repair, replication, and cell division[5,6]. In the absence of DNA damage, the bacterial LexA repressor binds to operator sequences within the promoter of SOS response genes (SOS boxes/Cheo boxes in *Bacillus subtilis*)[7,8]. LexA binds to these boxes as a dimer with each monomer comprising a C-terminal dimerisation domain (CTD) and an N-terminal DNA-binding domain (NTD)[9]. DNA damage results in the formation of single-stranded DNA fragments to which RecA binds resulting in an activated RecA-nucleoprotein filaments (RecA*)[10]. RecA* in turn promotes the auto-catalytic cleavage of unbound LexA[11], reducing the cellular LexA pool

[1]School of Infection & Immunity, University of Glasgow, G12 8TA Glasgow, UK. [2]Department of Biology, Faculty of Science, Al-Baha University, Al-Baha city, Al Aqiq 65779, Kingdom of Saudi Arabia. [3]MRC Centre for Molecular Bacteriology and Infection, Imperial College London, London SW7 2AZ, UK. [4]School of Medicine, University of St Andrews, North Haugh, St Andrews KY16 9TF, UK. ✉e-mail: afh22@st-andrews.ac.uk

and allowing the progressive activation of SOS-genes according to their LexA binding affinities. Cleavage of the LexA repressor separates its CTD and NTD, revealing latent ClpX-recognition motifs leading to the degradation of both fragments by the ClpXP protease complex[12]. Even though both fragments of the *E. coli* LexA protein are recognised by ClpX, ClpX is mainly involved in the degradation of the LexA NTD in vivo[12], while the Lon protease is responsible for degrading the LexA CTD[13]. In *Staphylococcus aureus*, the ATPases ClpX and ClpC along with the proteolytic activity of ClpP are involved in the degradation of the LexA NTD and subsequent SOS activation[14]. Since the LexA NTD retains some of its ability to bind to SOS boxes and to repress target genes in both Gram-positive and Gram-negative bacteria[14–16], its targeted removal by ClpXP is required for SOS activation.

Prophage induction is a fundamental process in phage biology and is controlled by the CI protein in phage λ. λ CI is a LexA-like repressor comprised of an NTD involved in DNA binding and repression and a CTD that upon recognition of activated RecA* are autocatalytically separated[17]. CI repressors are present in a wide variety of temperate phage genomes of both Gram-positive and Gram-negative hosts[18,19]. Interestingly, several examples of phage CI repressor molecules have been identified that retain some and sometimes even full binding affinities with their cognate promoter DNA after autocatalytic separation of their C- and NTDs[20–23]. The retention of such levels of DNA affinity by the CI NTD therefore would require its active inactivation to ensure progression from the lysogenic to the lytic phage lifecycle. Despite its vital role in temperate phage biology, how the CI NTD repressor activity is abolished after RecA* mediated release remains elusive.

The proteolytic degradation of the LexA NTD together with the conserved domain architecture found in both LexA and LexA-like repressors such as CI suggest that the bacterial proteasome and specifically Clp proteases may also be involved in prophage induction. These proteases mediate ATP-dependent protein degradation and consist of a proteolytic subunit (ClpP) associated with an AAA-ATPase (ClpX, ClpA, ClpC, and ClpE with ClpP; ClpY/HslU with ClpQ/HslV)[24]. ClpP alone is only able to degrade small peptides and requires association with one of its ATPases which act as energy-dependent unfoldases, unfolding and delivering the protein into the ClpP proteolytic chamber[25]. The target specificity of the protease complex is determined by its ATPase, which recognises specific degron motifs, often with the assistance of additional adaptor proteins[24,26–28].

Here we show that the ClpX ATPase performs an essential role in prophage induction by abolishing the ability of the CI NTD to repress lytic phage genes after RecA* mediated CI cleavage and allowing the prophage to enter lytic replication. This process is distinct from the role of ClpX on the staphylococcal SOS response and acts via direct binding of ClpX to the CI NTD. By contrast, the ClpP protease and its proteolytic interaction with ClpX are not required for this process but are essential for the activation of the staphylococcal SOS response.

## Results

### Clp proteases are not involved in the lytic phage cycle

We initiated this study testing the possibility that the chromosomally encoded Clp ATPases or proteolytic subunits (Table S1) could control the life cycles of the prototypical *S. aureus* phages Φ11 and 80α, which have been extensively used as models for Gram-positive phages[29,30]. Both phages encode a CI-like repressor that maintains the phage in its lysogenic prophage state within the bacterial chromosome. To test whether any of the Clp proteases (B, C, L, P, Q, X or Y; Table S1) were involved in the reproduction of phages Φ11 and 80α, we initially evaluated the lytic cycle of these phages. Lysates of either phage Φ11 or 80α were used to infect strain RN450 or a collection of derivative mutant strains in which the different *clp* genes were either deleted or inactivated by the insertion of an erythromycin resistance cassette. No differences in either the number or size of the phage plaques

compared to the wt staphylococcal strain were observed when the different mutant strains were infected (Fig. S1). Since plaque formation requires normal phage replication, packaging and lysis, this result suggests that the Clp proteases were not required in these steps of the phage cycles.

### ClpP and ClpX are involved in prophage induction in *S. aureus*

To investigate the potential involvement of Clp proteases in the induction stage of the temperate phage cycle, we performed experiments using RN450 derivatives with deletions or disruptions of various *clp* genes. The strains were lysogenised with Φ11 or 80α phages, and the prophages were induced using MitC to activate the SOS response. Subsequently, we analysed the phage titres using strain RN4220 as recipient.

Most of the mutant strains, except for the *clp*X and *clp*P mutants, underwent lysis by 4 h after SOS induction, indicating the involvement of ClpX and ClpP in prophage induction. Specifically, the *clp*P mutant exhibited a reduction in phage titre of $10^5$–$10^6$-fold compared to the wt strain upon prophage induction (Φ11 and 80α, respectively). Similarly, the *clp*X mutant showed a significant reduction in phage tires ($10^8$–$10^9$-fold) and produced very few plaques (10–100 PFU per ml) (Fig. S1), highlighting the critical role of ClpX in prophage induction.

To confirm the previous results, we generated a new set of in-frame deletion mutants in which the *clp*X and *clp*P genes were deleted using allelic replacement (strains JP18031 and JP18030, respectively). These strains were either infected with Φ11 or 80α or lysogenised with these phages (JP18157: Φ11 Δ*clp*X; JP18158: Φ11 Δ*clp*P, JP18169: 80α Δ*clp*X; JP18170: 80α Δ*clp*P) and the phage cycle induced with MitC. In agreement with the previous results, lysis behaviour and phage titres remained unaffected during phage infection (Fig. 1a). After prophage induction, both the Δ*clp*P and Δ*clp*X mutants did not lyse after 4 h and only the Δ*clp*P mutant lysed after overnight incubation while the Δ*clp*X mutant lysates remained turbid (Fig. S2). Consistent with this, phage titres were reduced ~$10^8$/$10^9$-fold in the Δ*clp*X and ~$10^4$-fold in the Δ*clp*P mutant (Fig. 1b). Introducing plasmid-borne versions of either *clp*X (pJP2601, JP18381, or JP18919) or *clp*P (pJP2602, JP18383, or JP18921) into the respective mutants under the control of a cadmium-inducible promoter fully restored wild-type phage lysis and titres (Fig. 1c) confirming their role in prophage induction.

### ClpX and ClpP have different roles in SOS and prophage induction

Since prophage induction and activation of the bacterial SOS response are linked, and since in *S. aureus* the latter process is affected in the absence of the ClpXP proteases[14], we tested whether the impact of the *clp*X or *clp*P mutations on phage induction could be explained by their role on SOS response activation. This was unlikely, however, since the phenotypes of both mutants (almost complete inability or reduction of phage progeny production for the *clp*X and *clp*P mutants, respectively) were inconsistent with the known roles of ClpXP in SOS response induction[14], where the concerted action of ClpXP is required for LexA NTD degradation. To explore this further, we constructed a set of reporter plasmids in which the promoters of the SOS-inducible genes *lex*A and *rec*A were fused to a β-lactamase reporter gene in plasmid pCN41 (pJP2596 and pJP2597, respectively). These were introduced into the RN4220 wt, RN4220 Δ*clp*P and RN4220 Δ*clp*X mutant strains and the SOS response induced by MitC addition.

Consistent with their SOS-inducibility, both *lex*A and *rec*A promoters were activated upon MitC addition in the wt *S. aureus* strain background (JP20858 and JP20417, respectively) showing a 10.0- and 7.6-fold induction, respectively (Fig. 2). No induction of the *lex*A or *rec*A promoters (1.2- and 1.0-fold induction compared to uninduced, respectively) was observed in the Δ*clp*P mutant background (JP20860 and JP20419, respectively), showing that ClpP was essential for the activation of the SOS response. This confirmed that ClpP-mediated

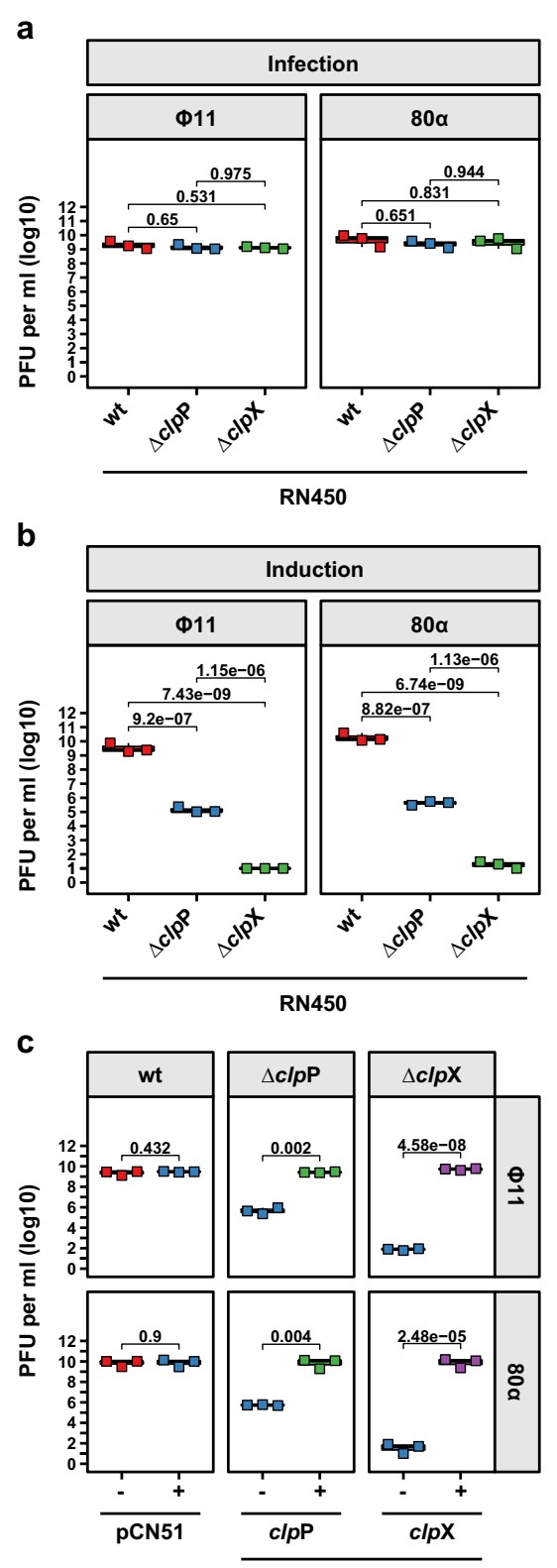

**a**

**Infection**

Φ11 / 80α

PFU per ml (log10)

RN450

**b**

**Induction**

Φ11 / 80α

PFU per ml (log10)

RN450

**c**

wt / Δ*clp*P / Δ*clp*X

PFU per ml (log10)

Φ11 / 80α

pCN51 / *clp*P / *clp*X

pCN51 expressing

**Fig. 1 | ClpP and ClpX are involved in phage induction but not phage infection.**
The defined RN450 derivative strains were either **a** infected with the indicated
phages or **b** lysogenic derivatives thereof induced by MitC addition. **c** The *clp*P and
*clp*X genes were cloned into the cadmium-inducible expression plasmid pCN51,
introduced into the defined strains, and induced by MitC addition. Expression from
the pCN51 plasmids was maintained throughout the experiment by the addition of
$1\,\mu$M $CdCl_2$. **a**–**c** Plaque formation was assessed on a lawn of RN4220. Bold hor-
izontal lines in each boxplot represent the median and lower and upper hinges of
the first and third quartiles, respectively ($n = 3$ biological replicates). Assessment of
statistically significant differences between groups was performed using ANOVA
followed by Tukey's HSD post-test **a**, **b** or a two-sided Student's *t* test **c** on $log_{10}$
transformed data. *p* values are indicated above the respective comparison.

and JP20418, respectively). As expected, neither reporter was inducible
by MitC addition and expression levels mimicked those observed in
the Δ*clp*P mutant background.

By contrast, when the *lex*A and *rec*A reporter plasmids were
introduced in the Δ*clp*X mutant background (JP20861 and JP20420,
respectively), although the overall transcriptional levels of both *lex*A
and *rec*A reporters were reduced (Fig. S3), they could still be induced
by MitC with comparable fold induction changes to the reporters in
the wt background (9.9- and 6.7-fold induction, respectively) (Fig. 2).
This suggested that ClpX was involved but not essential in inducing the
staphylococcal SOS response.

Taken together, the impact of the *clp*P and *clp*X mutation on
prophage induction and SOS response activation, as observed through
*lex*A and *rec*A expression, indicates that their roles in these processes
are distinct.

**ClpX is essential for prophage derepression**
Given the distinct roles of ClpP and ClpX in SOS- and prophage
induction, we aimed to investigate their specific functions in prophage
induction. We hypothesised that the reduced prophage titres
observed in the Δ*clp*P and Δ*clp*X mutants were unlikely to be caused by
failures in prophage replication, packaging, or phage release, as the
phage lytic cycle remained unaffected in these mutants. Furthermore,
staphylococcal prophages such as Φ11 or 80α, do not follow the
classical excision, replication, and packaging (ERP) cycle, where the
prophage first excises from the bacterial genome. Instead, they follow
the replication, packaging, and excision (RPE) cycle, where the
prophage initiates replication and packaging while still integrated into
the bacterial genome and excises late in the induction cycle[29]. Hence, it
was clear that ClpP and ClpX should control the heart of the lysogenic
switch system and should control prophage derepression.

To comprehensively assess the role of ClpP and ClpX in prophage
induction and the RPE cycle, we conducted a combination of quanti-
tative whole-genome sequencing[29] and Southern blotting experiments
before and after MitC induction of RN450, RN450 Δ*clp*X (JP18031) or
RN450 Δ*clp*P (JP18030) lysogenised with either Φ11 or 80α (Fig. 3 and
Fig. S4). The obtained sequencing reads were mapped to the respec-
tive strain's genome, normalised, and graphed (Fig. 3). The wt lysogens
(JP18269 for Φ11 and JP18270 for 80α) showed continuous sequencing
coverage across both bacterial genome and prophage DNA prior to
induction. Upon MitC induction of these strains, we observed a sharp
increase in read coverage for the prophage DNA sequence, indicating
replicating prophage. It is worth noting that the increased coverage of
genomic DNA on either side of the prophages resulted from in situ
replication generated by the induced prophages. While this was evi-
dent in Φ11 at the assessed timepoint, in situ replication of 80α was not
yet visible as it occurs later[29].

To confirm the level of prophage excision and replication, we
mapped reads spanning either the empty, bacterial attachment site
(*att*B, reflecting prophage excision events from the bacterial chro-
mosome), the prophage left attachment site (*att*L, monitoring
prophage integration) or the phage attachment site (*att*P, measuring

degradation of the bacterial LexA NTD was required for SOS
induction[14].

To further validate the essential role of ClpP in SOS response
activation, we introduced the *lex*A and *rec*A reporter plasmids into an
RN4220 derivative strain expressing an SOS-insensitive version of the
LexA repressor (LexA$_{G94E}$) that can no longer catalyse autocleavage
and consequently can no longer induce the SOS response[31,32] (JP20859

lytic phage replication) to the appropriate reference genomes or circular prophage sequences. MitC induction of the wt strain background reduced the proportion of bacterial chromosomes lysogenised with

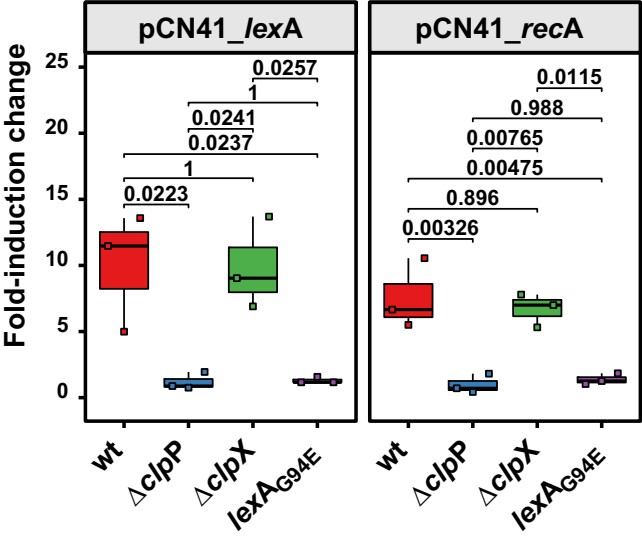

either prophage (Table 1). Moreover, it resulted in a 948-fold increase in circular prophage (*att*P) reads for Φ11 and a 257-fold increase for 80α (Table 1). These findings align with the results from Southern blotting experiments (Fig. S4), confirming prophage excision and replication in the wt background.

By contrast, the Δ*clp*P mutants (JP18158 for Φ11 and JP18170 for 80α) showed increased levels of circular, excised phage DNA compared to the wt strain prior to MitC induction (-13.2-fold for Φ11 and 3.4-fold for 80α) (Fig. 3, Table 1 & Source data). These results are consistent with the Southern blot data (Fig. S4), where a faint band is visible even in the absence of MitC induction. The amount of excised phage DNA in the Δ*clp*P mutant strains 60 min after MitC induction increased by 1.7-fold for Φ11 and remained close to the level observed in the uninduced sample for 80α (0.76-fold) (Fig. 3 & Table 1). However, reads spanning *att*P in the Δ*clp*P mutant background did not reach induction levels comparable to those observed in the wt background in line with the delayed lysis phenotype (Fig. S2). Nonetheless, Southern blotting experiments confirmed an increase in phage band intensity at later time points (Fig. S4, 90 and 120 min), further highlighting delayed prophage induction in the Δ*clp*P mutant background. It is important to note that no excision of the prophage from the chromosome could be mapped to the 80α *att*B site (0% of reads for both uninduced and induced samples), indicating full integration of the prophage in all bacterial genomes. Similar observations were made for the Δ*clp*P mutant Φ11 lysogen, where only 1.8% (uninduced sample) or 3.5% (induced sample) of reads were mapped across the *att*B sites.

Importantly, in the Δ*clp*X mutant lysogens, read coverage remained relatively constant and did not increase substantially after MitC induction (1.67-fold for Φ11 [JP18157] and 1.13-fold for 80α [JP18169] compared to uninduced), indicating no excision of the prophage (Fig. 3). Mapping across the *att*B and *att*L sites confirmed that the relevant reads mapped almost exclusively to the *att*L sites (100 and 100% for lysogens of 80α and 100 and 97.5% for Φ11 lysogens either without or with MitC induction, respectively). Interestingly, we still observed some reads mapping to the excised phage *att*P sites indicating that some excision was still possible in these strains. However, the total number of normalised reads in the Δ*clp*X mutant lysogens mapping to the *att*P sites was similar to the uninduced wt in the

**Fig. 2 | Distinct roles for ClpP and ClpX in SOS response induction.** Reporter plasmids were designed to place the β-lactamase reporter gene (*bla*Z) of plasmid pCN41 under the control of the SOS-controlled promoters of *lex*A or *rec*A. RN4220 derivative strains containing the indicated plasmids were grown to exponential phase, split and the SOS response was induced in one half of the culture with MitC while the other half was left untreated. Samples were taken 90 min after induction. Fold induction change of MitC-induced against non-induced samples is shown. Bold horizontal lines in each boxplot represent the median and lower and upper hinges of the first and third quartiles, respectively (*n* = 3 biological replicates). Assessment of statistically significant differences between groups was performed using ANOVA followed by Tukey's HSD post-test. *p* values are indicated above each comparison.

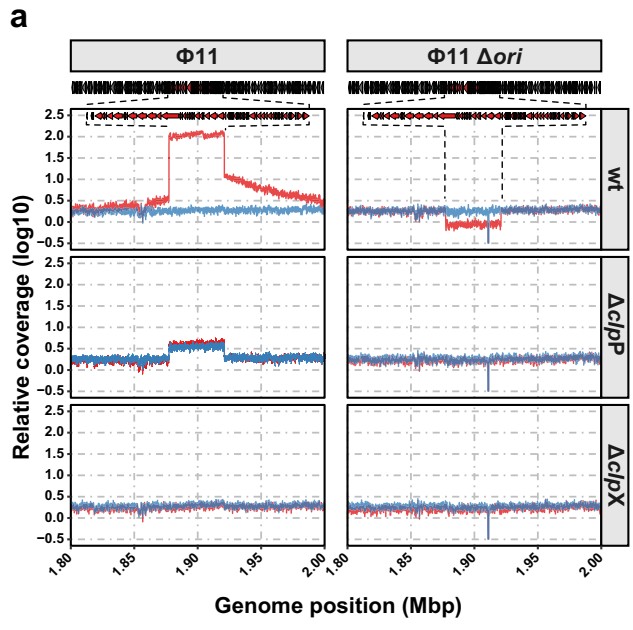
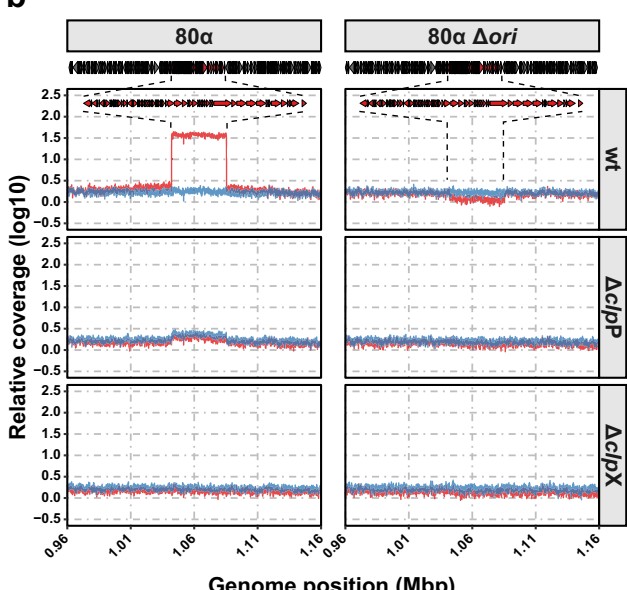

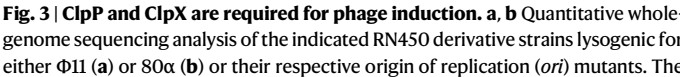

**Fig. 3 | ClpP and ClpX are required for phage induction. a, b** Quantitative whole-genome sequencing analysis of the indicated RN450 derivative strains lysogenic for either Φ11 (**a**) or 80α (**b**) or their respective origin of replication (*ori*) mutants. The indicated strains were induced with MitC and DNA was isolated either before (blue) or after 1 hour of induction (red) with MitC. DNA was then subjected to whole-genome sequencing and mapped to the relevant bacterial genomes.

**Table 1 | Fraction of integrated and excised prophages**

| | | Integrated (attL[a]) | | Excised (attP[b]) | |
|---|---|---|---|---|---|
| | | Uninduced | Induced | Uninduced | Induced |
| Φ11 | wt | 1.000 | 0.813 | 0.073 | 0.923 |
| | ΔclpP | 0.982 | 0.965 | 0.538 | 0.596 |
| | ΔclpX | 1.000 | 0.975 | 0.107 | 0.177 |
| Φ11 Δori | wt | 0.990 | 0.211 | 0.000 | 0.513 |
| | ΔclpP | 0.932 | 0.988 | 0.021 | 0.018 |
| | ΔclpX | 1.000 | 1.000 | 0.000 | 0.000 |
| 80α | wt | 0.996 | 0.697 | 0.079 | 0.967 |
| | ΔclpP | 1.000 | 1.000 | 0.235 | 0.227 |
| | ΔclpX | 1.000 | 1.000 | 0.022 | 0.036 |
| 80α Δori | wt | 0.991 | 0.639 | 0.005 | 0.076 |
| | ΔclpP | 0.968 | 0.985 | 0.024 | 0.015 |
| | ΔclpX | 0.985 | 1.000 | 0.007 | 0.031 |

[a]Fraction calculated as reads attL/(attB+attL). [b]Fraction calculated as reads attP/(attL+attP).

case of Φ11 and around 30% for 80α lysogens, always lower than those observed for either phage in the ΔclpP mutant lysogens. This suggests the possibility of spontaneous prophage induction, which will be further investigated later.

Contrary to the induced Φ11 lysogen in the wt background, where we observed a 6.2-fold increase in attL read coverage compared to the uninduced strain, indicating in situ replication of the prophage prior to excision[29], we did not observe any notable increase in attL coverage in either the ΔclpP or ΔclpX mutant Φ11 lysogens. Since endogenous replication does occur independently of prophage excision but requires prophage induction, this provided additional support for both ClpP and ClpX acting prior to replication and excision from the chromosome. Unfortunately, we were unable to verify this observation in the 80α lysogen backgrounds due to the slower kinetics of induction and endogenous replication, which were not covered in these experiments[29].

To fully confirm whether ClpP and ClpX targeted the core repression module rather than replication, packaging and excision, we also included two phage mutants in Φ11 and 80α defective in their origins of replication (ori) as controls for phages capable of excision but not replication and generated ΔclpX and ΔclpP mutants in these strain backgrounds. If ClpX and/or ClpP were involved in prophage induction (the removal and degradation of the CI repressor) or excision (excisionase is only expressed after CI degradation[29,33]), these mutants would not show any changes in prophage sequence coverage before and after SOS induction in the ori mutant lysogens. Conversely, if either ClpX or ClpP were involved in phage replication, sequence coverage in the mutant strains in the ori mutant lysogens would resemble the behaviour of the ori mutants in the wt RN450 strain background, and MitC induction would lead to the loss of sequence coverage.

In the wt RN450 strain background lysogenic for the Φ11 and 80α ori mutants (JP20045 and JP20046, respectively), we observed an appreciable drop in read coverage in the prophage region of the induced ori mutants in both Φ11 (2.6-fold, ratio induced to uninduced of 0.385) and 80α (1.6-fold, ratio induced to uninduced of 0.641) (Fig. 3), indicating their inability to replicate once excised from the chromosome. Consistent with this, reads mapping to the attL sites reduced after MitC induction of the lysogens in the wt strain background (Table 1), indicating the excision could occur in these mutant phages. Interestingly, although starting at similar levels, the proportion of reads mapping to the attL site for the Φ11 Δori mutant in the wt strain background decreased substantially more (21.1% compared to 81.3% for the wt phage), potentially indicating differences in the ability of the phage to reintegrate into the bacterial chromosome after

excision. This difference was less evident in the 80α Δori mutant (63.96% compared to 69.7% for the wt strain) and might reflect differences in the speed of 80α derepression and/or initial replication compared to Φ11.

Deletion of the ori-containing gene in either Φ11 or 80α in the ΔclpP mutant background (JP20189 and JP20192, respectively) eliminated the increased read coverage for the prophage observed with the ΔclpP single mutant prior to MitC induction. Furthermore, no difference in attL site coverage was observed after MitC induction with over 93 and 96% of the bacterial chromosomes being lysogenic for Φ11 or 80α, respectively (Fig. 3 and Table 1). These findings confirm that ClpP acts upstream of in situ replication and prophage excision. Given that ClpP is essential for SOS induction (Fig. 2), this likely dampens downstream prophage induction. However, this result does not exclude that ClpP might also be involved in the removal of CI. Similarly, deletion of the ori in the ΔclpX mutant background lysogens (JP20191 and JP20190, respectively) showed no differences in sequence coverage, with or without MitC induction (Fig. 3), and more than 98% of the bacterial population remained lysogenic for either phage (Table 1, attL). Together, these data provide strong evidence that both ClpP and ClpX are required for prophage induction and act prior to prophage in situ replication and excision.

## ClpX is required for the degradation of the CI N-terminal cleavage fragment

The results presented in Figs. 2 and 3 revealed that ClpX and ClpP have distinct roles in SOS and prophage induction. While both ClpX and ClpP were shown to affect prophage derepression, ClpX performed different and/or additional roles compared to ClpP. As the degradation of the phage repressor is the initial step in prophage induction, we hypothesised that ClpX and/or ClpP are likely involved in this process, potentially occurring after the initial RecA*-mediated cleavage of CI. Since it was not possible to generate phage mutants that expressed only a post-cleavage NTD of CI, we assessed the role of ClpX and ClpP using a set of reporter plasmids reconstituting the regulatory module of Φ11 in plasmid pCN41 in which the cro promoter was fused to a β-lactamase reporter gene (see schematic Fig. 4). These plasmids contained different versions of the CI repressor: (i) a wild-type CI (CI$_{wt}$, pJP2578), (ii) a non-degradable version (CI$_{G131E}$, pJP2590) insensitive to SOS induction[32,34], and (iii) a construct expressing only the post-cleavage N-terminal domain of CI (CI$_{G131*}$, pJP2589). This last construct mimics the N-terminal fragment of CI after SOS-induced cleavage and allowed us to study the ability of this fragment to block phage induction as well as to assess the impact of the ΔclpX (JP20999) and ΔclpP (JP19795) deletions on its processing.

We introduced these reporter plasmids into the wt RN4220 strain and monitored the expression of the β-lactamase reporter gene with or without MitC induction. The CI$_{wt}$ reporter plasmid (JP19910) repressed reporter gene expression, which increased by approximately 36-fold upon MitC addition (Fig. 4), confirming the SOS-inducibility of this construct. By contrast, the reporter plasmid expressing the non-cleavable CI$_{G131E}$ variant (JP19912) showed no induction by MitC, indicating its SOS insensitivity. Interestingly, the CI$_{G131*}$ reporter plasmid (JP19911) displayed higher expression levels even in the absence of MitC (16–18-fold higher than the uninduced wt construct without and with MitC, respectively), suggesting that the N-terminal domain of CI alone was insufficient to repress reporter expression in the wt strain background.

Next, we introduced the reporter plasmids into the ΔclpP mutant background and repeated the experiment. The ΔclpP mutant carrying either the CI$_{wt}$ reporter plasmid (JP19918) or the CI$_{G131E}$ reporter plasmid (JP19920) showed no expression either with or without MitC induction. However, the CI$_{G131*}$ reporter plasmid in the ΔclpP background (JP19919) exhibited high β-lactamase expression regardless of MitC presence (Fig. 4). Notably, in the absence of MitC, the reporter

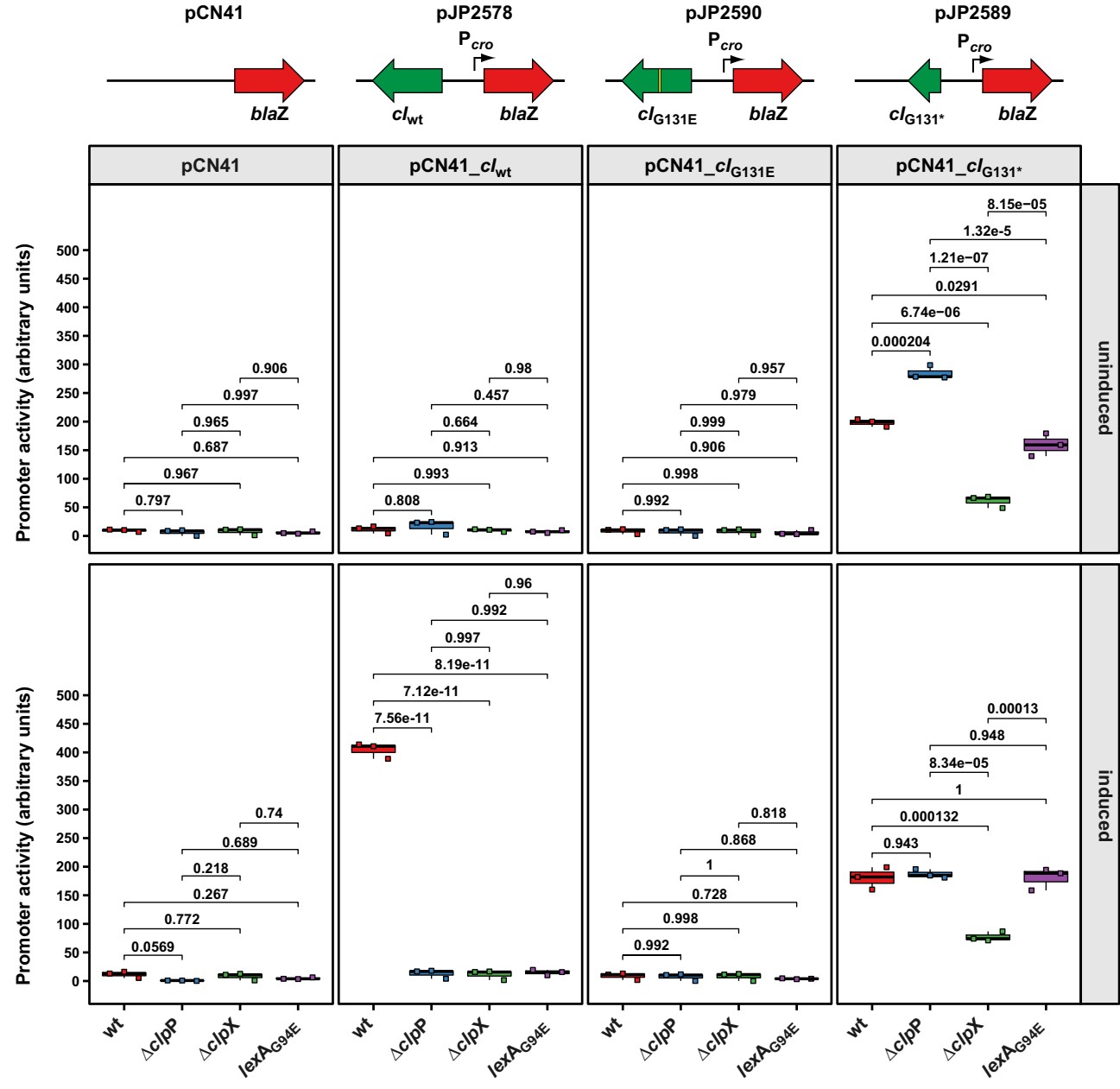

**Fig. 4 | Distinct roles for ClpP and ClpX in phage induction.** Plasmid pCN41-derived reporter plasmids were designed to place the β-lactamase reporter gene (*bla*Z) under the control of the Φ11 *cro* promoter. These plasmids also contained the genes encoding for either the Φ11 WT CI (*cl*$_{WT}$), an SOS-insensitive CI mutant (*cl*$_{G131E}$) or the post-cleavage N-terminal domain of CI alone (*cl*$_{G131*}$). Strains containing the indicated plasmids were grown to exponential phase, split and the SOS response induced in one-half of the culture with MitC. Samples were taken 90 min after induction. Bold horizontal lines in each boxplot represent the median and lower and upper hinges of the first and third quartiles, respectively ($n = 3$ biological replicates). Assessment of statistically significant differences between groups was performed using ANOVA followed by Tukey's HSD post-test. *p* values are indicated above each comparison.

expression in this strain background was 45.5-fold higher than that observed in the uninduced CI$_{wt}$ reporter plasmid, while MitC induction reduced this to 29.9-fold. These data strongly suggest that ClpP, in contrast to its role in SOS induction, is not required for the removal of the N-terminal fragment of CI.

When we introduced the reporter plasmids into the Δ*clp*X mutant background, we found that both the plasmid expressing CI$_{wt}$ (JP19914) as well as the plasmid expressing CI$_{G131E}$ (JP19916) were fully repressed either with or without the addition of MitC, showing only background levels of reporter gene expression (Fig. 4). Interestingly, the reporter plasmid expressing the NTD post-cleavage fragment CI$_{G131*}$ (JP19915)

was repressed in the Δ*clp*X mutant background and this repression could not be relieved by MitC addition (Fig. 4) (9.8- and 12.4-fold compared to the uninduced reporter plasmid containing the CI$_{wt}$ gene) confirming that (i) the CI$_{G131*}$ NTD post-cleavage fragment was still able to bind and repress the *cro* promoter and that (ii) ClpX was required for abrogating this repression.

Both ClpX and ClpP are also involved in the initial activation of the bacterial SOS response. To determine the SOS-independent contribution of ClpX and ClpP on the processing and degradation of CI, we introduce all reporter plasmids into a strain expressing an SOS-insensitive LexA$_{G94E}$ protein (JP1841). This mutant is no longer able to

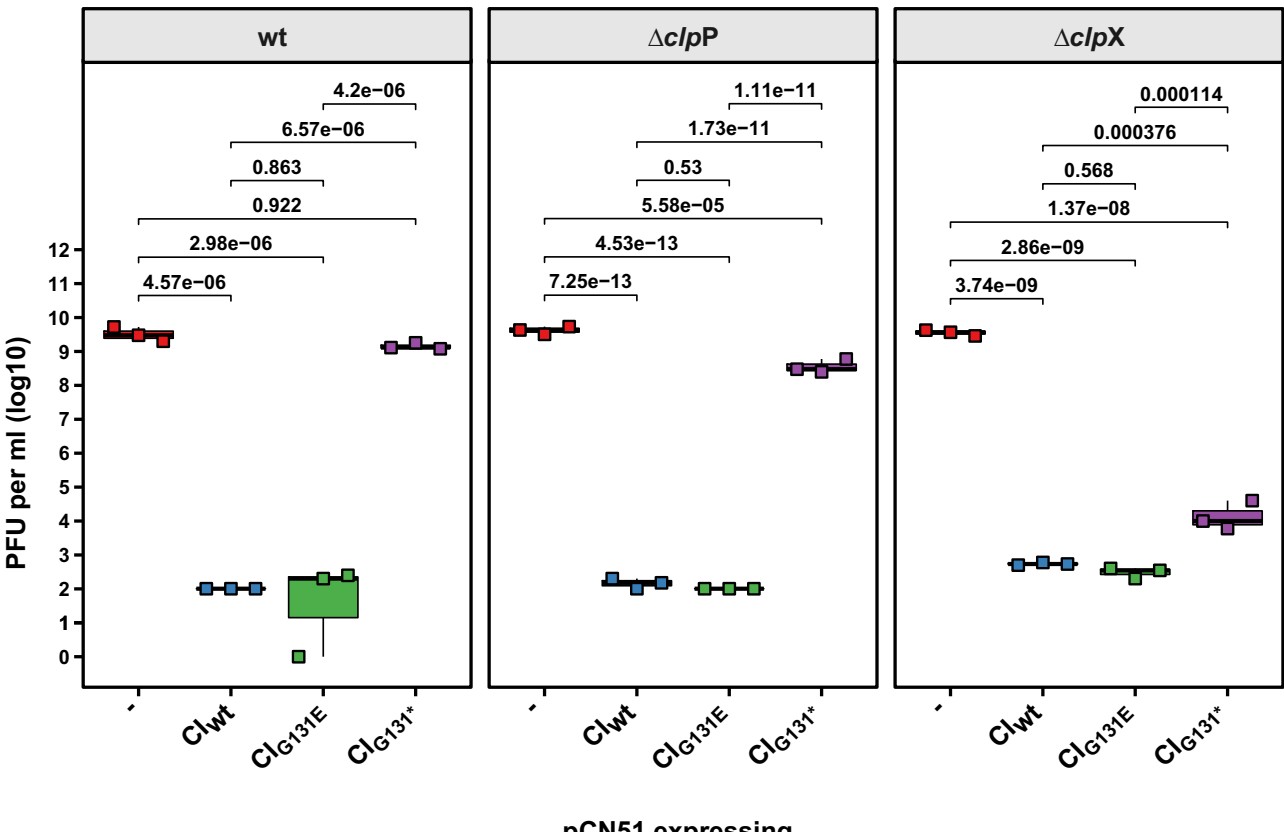

**Fig. 5 | The Φ11 CI N-terminal domain affects phage infection in the absence of either ClpP or ClpX.** Genes encoding either the Φ11 CI WT ($CI_{WT}$), an SOS-insensitive version ($CI_{G131E}$) or only its post-cleavage N-terminal fragment ($CI_{G131*}$) were cloned into the inducible expression plasmid pCN51 and introduced into the indicated strains. Expression from these plasmids was maintained throughout the experiment by the addition of 1 μM $CdCl_2$ during both growth and phage titration. Lawns of the defined, exponential phage, RN450 derivative strains were prepared on PB plates supplemented with 1 μM $CdCl_2$ to maintain CI expression and serial dilutions of Φ11 lysate spotted onto these lawns. Bold horizontal lines in each boxplot represent the median and lower and upper hinges of the first and third quartiles, respectively ($n = 3$ biological replicates). Assessment of statistically significant differences between groups was performed using ANOVA followed by Tukey's HSD post-test. $p$ values are indicated above each comparison.

induce the bacterial SOS response, particularly its further amplification through derepression of RecA. RecA levels and consequently RecA* levels therefore remain under the threshold required for the initial RecA*-mediated CI cleavage step in prophage activation. As expected, the $CI_{wt}$-expressing plasmid could no longer be induced by MitC induction in this strain background (JP20406), confirming that autocatalytic cleavage of CI could no longer be triggered by RecA* present in this background. Logically, this was also the case when introducing the SOS-insensitive $CI_{G131E}$ reporter into the $LexA_{G94E}$ strain background (JP20407). However, no changes compared to the wt strain were observed with the $CI_{G131*}$ reporter plasmid (JP20408) (Fig. 4) confirming that degradation of the $CI_{G131*}$ fragment occurred independently of any upstream SOS induction-related processes. These data therefore strongly indicated that only ClpX was essential for the processing of the CI NTD after SOS induction, while ClpP likely acted upstream through its role in SOS induction.

**The CI-NTD blocks phage infection only in the absence of ClpX**
To further validate these observations, we exploited the intrinsic resistance of phage lysogens to superinfection by another phage of the same repressor type. Expression of the CI repressor from the prophage can effectively block the replication of incoming phages in a process called superinfection immunity[35]. We reasoned that the NTD of CI alone would be insufficient to block phage superinfection, particularly in the presence of ClpX, which would lead to its elimination. However, the absence of proteins involved in the processing of the CI-NTD would

result in the accumulation of the repressor fragment, thereby reducing phage titres after infection.

To test this, we cloned the different $cI$ alleles of Φ11 ($cI_{wt}$, $cI_{G131E}$ and $cI_{G131*}$) into the cadmium-inducible expression vector pCN51 (pJP2584, pJP2585 and pJP2586, respectively). These plasmids were introduced into either the RN450 wt, its $\Delta clp$X or $\Delta clp$P mutant derivatives and the ability of Φ11 to infect these strains was assessed (Fig. 5). In the presence of the pCN51 control plasmid, the wt (JP19532), $\Delta clp$P (JP19540) or $\Delta clp$X (JP19536) mutant strains were equally susceptible to infection with Φ11 (Fig. 5). However, when the wt RN450 strain expressed either $CI_{wt}$ (JP19529) or the non-degradable $CI_{G131E}$ (JP19530), phage infection was almost completely blocked, resulting in a ~7-log reduction in phage titres. By contrast, the overexpression of the $CI_{G131*}$ NTD (JP19531) in the wt strain remained susceptible to phage superinfection comparable to the empty plasmid control strain.

In the $\Delta clp$P mutant background, the overexpression of $CI_{wt}$ (JP19537) and $CI_{G131E}$ (JP19538) resulted in complete blockage of Φ11 superinfection (~7-log reduction), while the overexpression of $CI_{G131*}$ (JP19539) led to a 10-fold reduction in phage titres (Fig. 5). Similarly, in the $\Delta clp$X mutant background, the overexpression of $CI_{wt}$ (JP19533) and $CI_{G131E}$ (JP19534) effectively blocked superinfection (~7-log reduction). Interestingly, consistent with the central role of ClpX in the removal of the CI-NTD, the overexpression of $CI_{G131*}$ in the $\Delta clp$X mutant background (JP19535) resulted in a reduction in phage titres by more than 5-log units (Fig. 5). Taken together, these data confirmed

that ClpX was responsible for inactivating the ability of the CI-NTD repressor fragment to block phage infection and replication, while ClpP only played a minor role in the turnover of the CI-NTD.

## ClpX-dependent prophage induction does not require interaction with ClpP

ClpX functions both as a substrate specificity protein, directing target proteins to the ClpP protease, and as a chaperone, assisting in correct protein folding[36]. Previous studies have shown that the interaction between ClpX and ClpP is dependent on a single amino acid and can be disrupted by introducing the I265E substitution into ClpX[37]. This mutant variant retains its chaperone function but loses the ability to facilitate proteolytic cleavage of target proteins. To investigate whether the interaction between ClpX and ClpP is necessary for ClpX's role in phage induction, we cloned a *clp*X gene encoding either the wild-type form (ClpX$_{wt}$) or the I265E substitution variant (ClpX$_{I265E}$) into the cadmium-inducible expression vector pCN51 (pJP2601 and pJP2605, respectively). These plasmids were then introduced into the Δ*clp*X mutant lysogens of Φ11 (JP18381 and JP21189, respectively) and 80α (JP18919 and JP21190, respectively).

While expression of the wt ClpX protein (ClpX$_{wt}$) in the Δ*clp*X mutant fully restored wt phage titres, expression of ClpX$_{I265E}$ also restored phage titres to 1.5-logs below wt levels (Fig. 6a), indicating that proteolytic complex formation of ClpX and ClpP per se was not required for prophage induction. Consistent with these results, expression of ClpX$_{I265E}$ in the Δ*clp*X mutant background partially restored phage replication (Fig. 6b). However, it is worth noting that a faint band resembling the one observed in the uninduced Δ*clp*P mutant appeared in the uninduced Δ*clp*X mutant strain expressing ClpX$_{I265E}$, particularly noticeable in the 80α sample. This suggests that there may be additional cellular processes influenced by the ClpX and ClpP interaction, which could impact normal phage replication in this specific strain background.

## ClpX binds to the CI N-terminal fragment but not the full-length CI repressor

Our previous findings demonstrated that the CI-NTD cleavage fragment alone retained repressor functionality and could block prophage induction and infection in the absence of ClpX. Furthermore, ClpX, but not ClpP, was essential for relieving this repression, suggesting that ClpX could interact with the N-terminal CI fragment but not the full-length protein. To investigate this further, we used a bacterial two-hybrid system, where we cloned CI$_{wt}$ or CI$_{G131*}$ as C-terminal fusion products to the T25 fragment of the *Bordetella pertussis* adenylate cyclase in plasmid pKT25 (pJP2636 and pJP2632, respectively) and the ClpX$_{wt}$ or ClpX$_{I265E}$ mutant as N-terminal fusion product to the T18 fragment of the *B. pertussis* adenylate cyclase in plasmid pUT18c (pJP2642 and pJP2638, respectively, see schematic Fig. 6c). While ClpX did not substantially interact with the full-length CI$_{wt}$, it interacted more strongly with the CI$_{G131*}$ construct expressing only the CI-NTD (Fig. 6c), confirming that ClpX acted after SOS-triggered CI-auto-cleavage. Notably, the ClpX$_{I265E}$ mutant, lacking the ability to interact with ClpP, displayed an interaction pattern identical to that of ClpX$_{wt}$, binding exclusively to the separated CI-NTD, CI$_{G131*}$ (Fig. 6c). Collectively, these data provide compelling evidence that ClpX selectively binds to the CI N-terminal fragment (CI$_{G131*}$) rather than the full-length CI$_{wt}$ protein. Moreover, the binding of ClpX to the N-terminal fragment alone is sufficient to trigger phage induction, independent of ClpP-mediated proteolytic degradation.

## Loss of ClpP and ClpX affects spontaneous prophage release

The results presented in Fig. 3 and S4 demonstrated that the absence of ClpP resulted in a low level of phage replication. Previous studies have shown that even in strains insensitive to the SOS response due to *rec*A mutations or the expression of an insensitive LexA protein, there

is always some basal induction of resident prophages[38–41]. Hence, we sought to investigate whether the Δ*clp*P and/or Δ*clp*X mutants had any influence on spontaneous phage release. To assess this, we cultured the wt, Δ*clp*P, and/or Δ*clp*X mutants lysogenic for Φ11 or 80α, following the same experimental procedure as with MitC induction but without adding MitC. The released phage titres were then determined by plating on a lawn of RN4220, and the results were normalised to the phage release from the wt background to account for inter-experimental variability.

Our findings revealed that the loss of ClpP led to a significant increase in phage release for both Φ11 (1.75-fold) and 80α (8.39-fold) lysogens (Fig. 7). This finding is consistent with the observed higher read coverage for the phages in these mutants. Conversely, the loss of ClpX resulted in a significant reduction in phage release, reducing the titres by 2.5- and 2.0-logs for Φ11 and 80α, respectively (Fig. 7). Notably, the recovered phage titres were identical to those observed in the samples induced with MitC, indicating that phages released after MitC induction in the Δ*clp*X mutants were the result of spontaneous, ClpX-independent induction events. Therefore, both ClpP and ClpX proteins exerted distinct effects on spontaneous prophage release, but with opposite effects.

## Discussion

Temperate bacteriophages can persist either in a lytic, replicative state or they can be integrated into their host's chromosome during their lysogenic lifecycle as prophages. During the lysogenic lifecycle, the prophage replication genes are repressed by a master repressor, in this case, CI. Activation of prophages allowing them to enter the lytic phage cycle is a fundamental process in biology that affects bacterial populations, horizontal gene transfer and evolution[42]. Despite its fundamental nature, the mechanisms by which CI is fully eliminated to alleviate prophage repression have not been comprehensively resolved to date. The CI repressor shares similarity in its domain architecture and processing to the LexA repressor of the SOS response (see Fig. S5). Inactivation of both repressors is facilitated initially through their interaction with activated RecA* caused by DNA damage[17,43] (Fig. 8). This results in the autocatalytic cleavage of the repressors and separates them into a DNA-binding NTD and a CTD required for repressor dimerisation. Importantly, both LexA[14–16] and CI[20–23] NTDs retain their ability to bind and repress their target promoters and require additional factors to alleviate repression (Fig. 8). Previous studies have shown that ClpX and ClpP were both involved in the removal of the NTD of the LexA SOS response repressor[12,14,28]. The removal of the NTD of the LexA repressor is crucial for the full activation of the SOS response as it retains the ability to bind and to repress SOS genes[14–16]. Here we show, for the first time, that the ClpX ATPase activates phage replication by eliminating the CI NTD ability to repress the prophage (Fig. 8). By contrast, the ClpP proteolytic subunit primarily acts through its essential role in SOS induction rather than CI repressor degradation in prophage induction. The divergent roles for ClpP in SOS- and prophage induction might be indicative of their role in bacterial physiology and the probabilistic fate of the bacterial cell. The SOS response system is triggered to save the bacterial host and slow down physiological processes while inducing DNA repair mechanisms[6]. ClpP-driven LexA NTD turnover might be a crucial component in returning the bacterial cell to normal growth once DNA damage has been resolved. By contrast, prophage induction occurs only after the cell has undergone substantial, potentially irreparable DNA damage as a consequence of a lower RecA* affinity to CI compared to LexA[43]. Thus, it might be evolutionary favourable for the phage to abandon its host under such conditions. Since this is fatal to the host, the phage would not gain any additional benefit by expending cellular resources.

ClpX specifically interacts with the CI-NTD after SOS-induced cleavage and not with the full-length CI proteins. Furthermore, the use

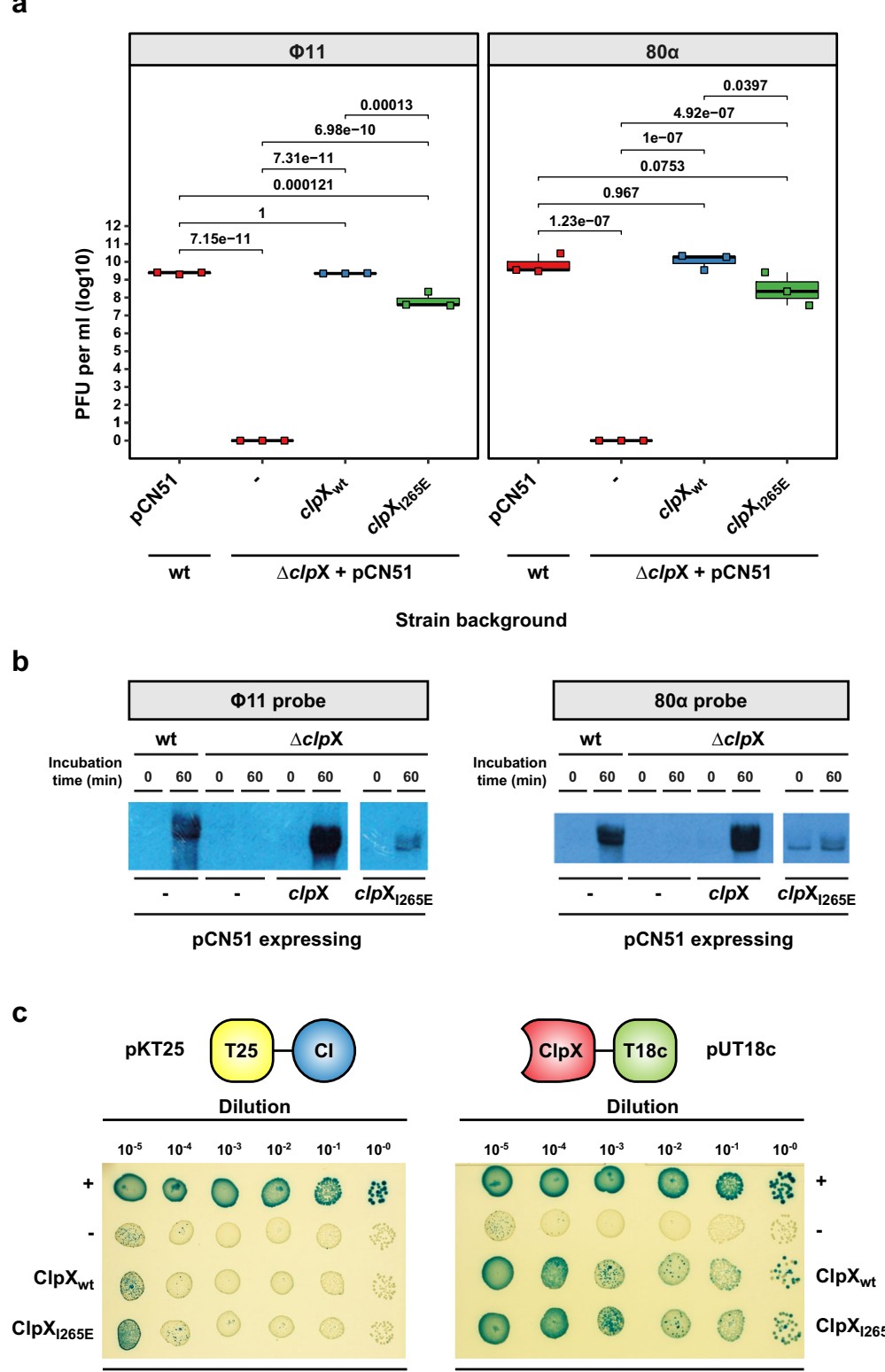

**a**

**b**

**c**

of a ClpX protein unable to shuttle substrates for proteolytic degradation but able to perform its chaperone function[37] confirmed that ClpX-dependent prophage induction was independent of the proteolytic degradation of the CI N-terminal fragment as opposed to its role in LexA NTD degradation[14]. This observation is consistent with the known ability of ClpX to bind and unfold certain protein substrates such as casein in the absence of ClpP[25] as well as the role of the

unfoldase function of ClpX in the lifecycle of several MGEs. For example, ClpX is known to interact with the phage Mu transposome complex, where its chaperone/unfoldase activity selectively destabilises the complex without causing its degradation[44]. Similarly, the ClpX unfoldase activity can activate the plasmid replication initiation factor TrfA[45]. While there is no role for ClpXP in the degradation of the lambda CI protein, several key regulatory proteins involved in the lysis/

**Fig. 6 | Interaction of ClpX and ClpP is not required for phage induction. a** The *clp*X wt gene (*clp*X$_{wt}$) or a *clp*X gene encoding a ClpX mutant unable to interact with ClpP (ClpX$_{I265E}$) were cloned into the inducible expression plasmid pCN51, introduced into either the RN4220 wt or its Δ*clp*X mutant derivative lysogenic for Φ11 or 80α and induced by MitC. Expression from these plasmids was maintained throughout the experiment by the addition of 1 μM CdCl$_2$. Plaque formation was assessed on a lawn of RN4220. Bold horizontal lines in each boxplot represent the median and lower and upper hinges of the first and third quartiles, respectively (*n* = 3 biological replicates). Assessment of statistically significant differences between groups was performed using ANOVA followed by Tukey's HSD post-test. **b** Samples of the same strains as in **a** were taken for DNA extraction at the time points indicated. Crude DNA lysates for Southern blotting analysis were then separated by agarose gel electrophoresis, transferred onto a nitrocellulose membrane, and replicating phage DNA visualised using a phage-specific DIG-

labelled DNA probe. **c** Bacterial Two-Hybrid assay of either the Φ11 full-length CI protein (CI$_{wt}$) or the post-cleavage CI N-terminal domain only (CI$_{G131*}$). The gene encoding either the Φ11 full-length CI protein (CI$_{wt}$) or **c** the post-cleavage CI N-terminal domain only (CI$_{G131*}$) were cloned into pKT25 (pJP2636 or pJP2632, respectively), while genes encoding either WT *clp*X (*clp*X$_{wt}$) or ClpX unable to interact with ClpP (*clp*X$_{I265E}$) were cloned into pUT18c (pJP2642, pJP2638, respectively). The pUT18c- and pKT25-derivative plasmids were co-transformed into *E. coli* strain BTH101 and a single colony selected. Serial dilutions of an overnight culture were plated onto LB supplemented with kanamycin (30 μg ml$^{-1}$), ampicillin (100 μg ml$^{-1}$), 100 μM isopropyl β-d–1-thiogalactopyrano-side (IPTG) and 20 μg ml$^{-1}$ X-gal. BTH101 transformed with pUT18c-zip and pKNT25-zip or pUT18c and pKT25 served as positive or negative controls for protein–protein interactions, respectively.

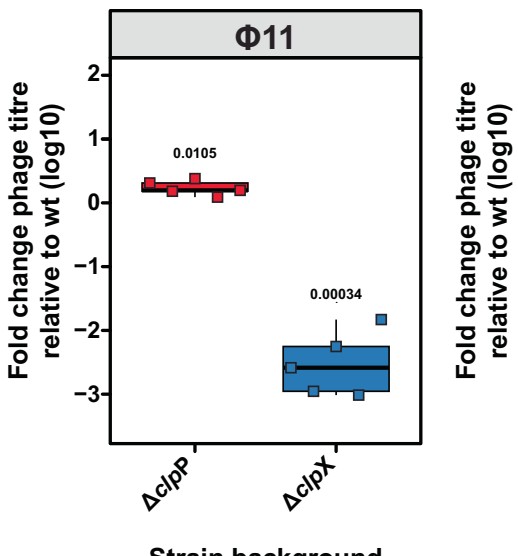
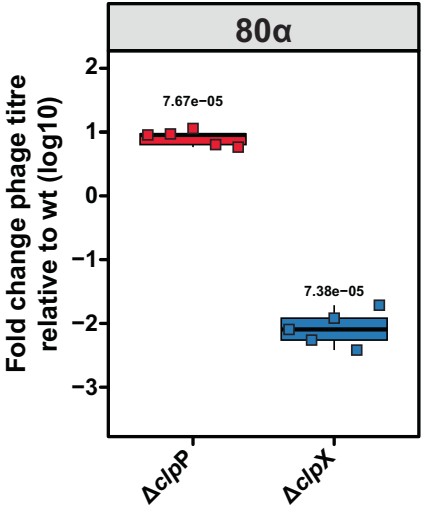

**Fig. 7 | ClpP and ClpX alter spontaneous prophage induction rates.** The indicated RN450 prophage lysogens were grown for MitC induction but were left untreated to monitor spontaneous prophage induction. Plaque formation was assessed on a lawn of RN4220 and normalised per experiment relative to phage titres in the wt strain background. Bold horizontal lines in each boxplot represent

the median and lower and upper hinges of the first and third quartiles, respectively (*n* = 3 biological replicates). Assessment of statistically significant differences between groups was performed using a two-sided Student's *t* test on log$_{10}$ transformed data assessing the hypothesis that phage titres were not different from wt (fold-change = 0). *p* values are indicated above the respective comparison.

lysogeny decision process in this model phage are subject to rapid proteolytic degradation. These are therefore rapidly lost once they are no longer synthesised. These instable proteins are O, N, CII, CIII and Xis and are degraded by ClpXP (O-protein), Lon (N-protein and Xis), and FtsH (CII and CIII)[4]. Presently, only the Mu repressor protein, which is different to LexA-like CI repressors, has been shown to be subject to ClpXP-mediated degradation. Even though the precise mechanism of this induction is still unknown, it is thought that a C-terminal ClpX-recognition motif that is present but conformationally inaccessible in the repressed state can be exposed by environmental triggers and thus lead to the degradation of the Mu repressor and subsequent activation of the Mu transposase[46–48]. The dual roles of the ClpX protein therefore necessitate the precise determination of whether the biological effects caused by ClpX are mediated via its chaperone or protein degradation function.

ClpP on the other hand was shown to be essential for the activation of the bacterial SOS response, and its absence consequently prevented full phage induction. Despite the inability of a *clp*P mutant to induce the SOS response, we still observed higher lysis, phage replication and higher phage titres compared to a *clp*X mutant, where phage replication was almost completely absent. Interestingly, loss of *clp*P gave rise to a faint phage replication band during Southern blot

analysis, to increased sequence coverage and *att*P circular phage site reads coverage during whole genome sequencing of the prophage region even in the absence of MitC induction. This further translated into an increased spontaneous release of phage particles without MitC induction and indicates that ClpP alone or in combination with other proteins performs (an) additional role(s) either required for full prophage stabilisation in the host's chromosome or in preventing erroneous induction events. This could be facilitated through interaction/degradation with/of other regulatory phage proteins modulating the lytic switch. Alternatively, loss of ClpP or loss of the ability of ClpX to interact with ClpP, could liberate additional ClpX which is able to interact with CI NTD fragments generated by self-processing at low rates[31]. Note that each ClpXP complex contains a heptameric ClpP and hexameric ClpX ring[25]. This would reduce the cellular pool of available full CI proteins and NTD-CI fragments that could repress the prophage. A reduction in the cellular repressor pool concentration could then result in an increased likelihood of prophage derepression in individual cells, and consequently produce elevated spontaneous phage release. This model is supported by the Southern blot phenotypes of the ClpX$_{I265E}$ mutant strain which mimics that of the Δ*clp*P mutant and produces a faint replicating phage band even without MitC induction (Fig. S4).

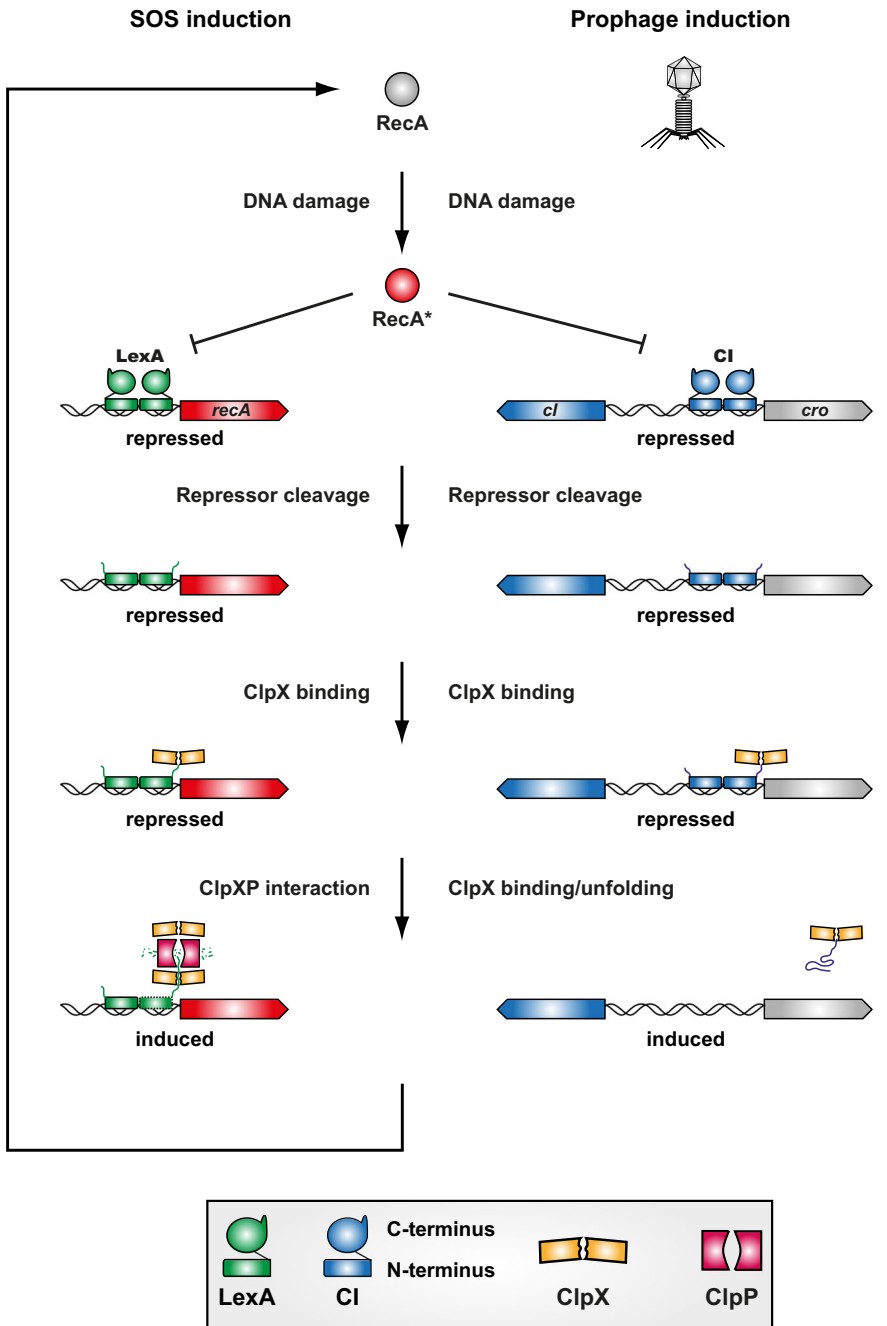

**Fig. 8 | Induction of the staphylococcal SOS response and prophages.** DNA damage activates the RecA (RecA*) protein which binds to both the LexA and the CI repressor catalysing their autocleavage. The repressor N-terminal domains retain some DNA-binding capacity. ClpX specifically binds to the N-terminal repressor domains after RecA* catalysed cleavage. For SOS induction, ClpX needs to interact with ClpP to facilitate the proteolytic degradation of the LexA N-terminus. This then also results in the increased expression of RecA further increasing SOS induction for as long as DNA damage is present. By contrast, prophage induction does not require the interaction of ClpX and ClpP and binding of ClpX to the N-terminal fragment of CI is sufficient for inducting the lytic phage cycle.

The slow prophage induction in the *clp*P mutant might be explained by basal RecA expression, which can be activated by DNA damage (RecA*). Thus, cumulatively, sufficient RecA* might be produced to result in a degree of prophage derepression despite an inability to induce the SOS response. By contrast, ClpX was important for spontaneous prophage release and its deletion resulted in a > 2-log reduction suggesting that its role in spontaneous and SOS-mediated prophage induction might be analogous. Because MitC-induced phage titres and spontaneously-induced phage titres in Δ*clp*X mutant were identical (Fig. S6), the released phage particles after MitC induction are likely the result of low levels of ClpX-independent spontaneous

prophage induction making ClpX essential for SOS-induced prophage induction.

In summary, we show that both ClpX and ClpP are involved in prophage induction in *S. aureus* (Fig. 8). Prior to SOS induction, CI is protected from the actions of ClpX, which only acts after SOS-mediated autocatalytic cleavage of CI by binding to the resulting NTD post-cleavage fragment and inactivating its DNA-binding capacity. While this function of ClpX alone is sufficient for prophage induction, ClpP might still be involved in degrading the CI N-terminal fragment to liberate ClpX, but this is not critical. Instead, ClpP primarily affects prophage induction via its essential role in the staphylococcal SOS

response. In addition to their role in SOS-mediated prophage induction, both ClpP and ClpX are also important for the levels of spontaneous prophage release. ClpX likely acts in a similar capacity in both SOS and spontaneous prophage induction as phage titres remained comparable either with or without MitC. By contrast, ClpP appears to perform additional regulatory functions controlling stable prophage integration in the bacterial chromosome or preventing erroneous induction events through regulating the levels of additional proteins involved in the lysogenic switch decision.

## Methods

### Bacterial strains and culture conditions
The bacterial strains used in this study are listed in Table S2. *S. aureus* strains were grown in Tryptic soy broth (TSB) at 37 °C and 120 rpm or on Tryptic soy agar (TSA) plates supplemented with sodium citrate (1.7 mM). *E. coli* strains were grown in Lysogeny broth (LB) 37 °C and 180 rpm and on LB agar plates. Antibiotic selection was used where appropriate (10 or 2.5 µg ml$^{-1}$ erythromycin (as indicated), 100 µg ml$^{-1}$ ampicillin and 30 µg ml$^{-1}$ kanamycin).

### Phage infection assay
Test strains were grown in TSB to the early exponential phase (OD$_{540}$-0.15) at 37 °C and 120 rpm. 12 ml of this culture were pelleted by centrifugation (3300 × $g$, 10 min) and the pellet was re-suspended in 6 ml of fresh TSB and 6 ml of phage buffer (1 mM MgSO$_4$, 4 mM CaCl$_2$, 50 mM Tris-Cl, 100 mM NaCl, pH = 8). Note that the phage buffer provides MgSO$_4$ and CaCl$_2$ required for phage adsorption. Equal numbers of phages (-10$^8$ PFU) were added to each culture and infected cultures were incubated for 4 h at 30 °C and 80 rpm followed by overnight incubation at room temperature. Cleared lysates were filtered using a 0.22 µm polyethersulfone syringe filter (Minisart, Sartorius).

### Phage induction and titration
*S. aureus* strains lysogenic for the required phages were grown to early exponential phase (OD$_{540}$-0.15) at 37 °C and 120 rpm. Cultures were then induced by the addition of mitomycin C (MitC) (2 µg ml$^{-1}$) and incubated for 4–5 h at 30 °C, 80 rpm followed by overnight incubation at room temperature before filtering using a 0.22 µm polyethersulfone syringe filter (Minisart, Sartorius). To determine the phage titres, RN4220 cultures were grown to OD$_{540}$-0.35 and 100 µl of this culture were mixed with 100 µl of serially diluted phage lysates in phage buffer (1 mM MgSO$_4$, 4 mM CaCl$_2$, 50 mM Tris-Cl, 100 mM NaCl, pH = 8). The mixtures were incubated for 5 min at room temperature, then 3 ml of molten phage top agar (PTA, 20 g l$^{-1}$ Nutrient Broth No. 2, Oxoid, plus 3.5 g l$^{-1}$ agar, Formedium supplemented with 10 mM CaCl$_2$, -50 °C) was added and the mixture overlaid onto phage base agar plates (20 g l$^{-1}$ Nutrient Broth No. 2, Oxoid, plus 7 g l$^{-1}$ agar, Formedium supplemented with 10 mM CaCl$_2$). Plates were incubated overnight at 37 °C and the plaque forming unit (PFU ml$^{-1}$) was determined.

### Lysogenisation of prophages
Recipient strains were grown to OD$_{540}$-0.35 and 100 µl of this culture were mixed with 3 ml of molten phage top agar (PTA, 20 g l$^{-1}$ Nutrient Broth No. 2, Oxoid, plus 3.5 g l$^{-1}$ agar, Formedium supplemented with 10 mM CaCl$_2$, -50 °C) and the mixture overlaid onto phage base agar plates (20 g l$^{-1}$ Nutrient Broth No. 2, Oxoid, plus 7 g l$^{-1}$ agar, Formedium supplemented with 10 mM CaCl$_2$). Serial dilutions of the prophage to be lysogenised in phage buffer (1 mM MgSO$_4$, 4 mM CaCl$_2$, 50 mM Tris-Cl, 100 mM NaCl, pH = 8) were spotted as 10 µl spots onto the recipient layer and dried. Plates were incubated overnight at 37 °C and a plaque forming unit (PFU ml$^{-1}$) was determined. Microcolonies of recipient strains in low dilutions of phage spots were selected and streaked onto fresh TSA plates supplemented with sodium citrate (1.7 mM) and antibiotics where required. Single colonies were restreaked several

times onto fresh plates and the presence of prophage lysogens was established by MitC induction (where possible) and PCR using primers Sa5-F and Sa5-R to confirm Φ11 lysogens and Sa7-F and Sa7-R for 80α lysogens[49].

### Superinfection immunity/efficiency of plating assay
*S. aureus* strains carrying the required plasmids were grown in TSB supplemented with the relevant antibiotics to an OD$_{540}$-0.35 and 100 µl of this culture were mixed with 3 ml of phage top agar (PTA, 20 g l$^{-1}$ Nutrient Broth No. 2, Oxoid, plus 3.5 g l$^{-1}$ agar, Formedium supplemented with 10 mM CaCl$_2$ and 1 µM CdCl$_2$) and overlaid onto phage base agar plates (20 g l$^{-1}$ Nutrient Broth No. 2, Oxoid, plus 7 g l$^{-1}$ agar, Formedium supplemented with 10 mM CaCl$_2$ and 1 µM CdCl$_2$). Phage lysates and dilutions in phage buffer (1 mM MgSO$_4$, 4 mM CaCl$_2$, 50 mM Tris-Cl, 100 mM NaCl, pH = 8) were spotted in triplicates of 10 µl each onto lawns of the specified strains, dried and incubated overnight at 37 °C prior to plaque forming unit (PFU ml$^{-1}$) determination.

### DNA manipulations
Plasmid constructs used in this study (Table S3) were generated by cloning PCR products (Kapa Hifi Polymerase, Roche) obtained with oligonucleotide primers listed in Table S4 and digested with the indicated restriction enzymes (New England Biolabs). Detection probes for phage DNA in Southern blots were generated by PCR using a non-proofreading polymerase (DreamTaq polymerase, ThermoFisher) using oligonucleotides specified in Table S4. Probe labelling and DNA hybridisation were performed following the protocol provided with the PCR-DIG DNA-labelling and chemiluminescent detection kit (Roche, catalogue number 11093657910).

### Generation of clean deletion mutants
The required clean deletion mutants were constructed using allelic replacement by cloning flanking regions up- and downstream of the respective gene (0.5–1 kbp) into pMAD[50] using oligonucleotide primers described in Table S4. The plasmids were then transformed into the required strains and integration of the plasmid was selected by growth at the restrictive temperature (42 °C) on TSA plates supplemented with 80 µg ml$^{-1}$ X-gal and 2.5 µg ml$^{-1}$ erythromycin. Single crossover events were isolated (light blue colonies) and grown overnight under replication-permissive conditions (TSB, 30 °C, 80 rpm) to facilitate excision and loss of the integrated plasmid. Serial dilutions of the cultures were plated on TSA plates supplemented with 80 µg ml$^{-1}$ X-gal and correct deletion mutants identified by PCR followed by sequencing using oligonucleotides annealing outside of the recombination region and indicated in Table S4.

### Southern blotting
Strains containing the defined phage and plasmids were grown to early exponential phase (OD$_{540}$-0.15) in 10 ml of TSB supplemented with antibiotics where plasmids were present. Phages were induced with MitC (2 µg ml$^{-1}$) and, where pCN51 expression plasmid derivatives were present, 1 µM CdCl$_2$ was added to induce expression as indicated. One ml samples were taken at the defined time points, pelleted by centrifugation (16873 × $g$, 2 min) and pellets shock frozen on dry ice. The pellets were re-suspended in 50 µl lysis buffer (47.5 µl TES-Sucrose (10 mM Tris-Cl, 100 mM NaCl, 1 mM EDTA, 20% (w/v) sucrose) and 2.5 µl lysostaphin [12.5 µg ml$^{-1}$]) and incubated at 37 °C for 1 h. Following the incubation, 55 µl of SDS 2% proteinase K buffer (47.25 µl H$_2$O, 5.25 µl SDS 20%, 2.5 µl proteinase K [20 mg ml$^{-1}$]) was added before incubation at 55 °C for 30 min. Samples were vortexed for 1 h with 11 µl of 10× loading dye followed by three cycles of 5 min incubations in dry ice/ethanol and at 65 °C in a water bath. Samples were run on 0.7% agarose gel at 25–30 V overnight. DNA was transferred by capillary action to a positively charged nylon membrane (Roche), processed as per the manufacturer's instructions, and exposed using a DIG-labelled probe (see

DNA methods) and anti-DIG antibody (1:10,000 (v/v), Roche, catalogue number 11093274910) before washing and visualisation.

## Two-hybrid assay

The two-hybrid assay for protein−protein interaction was performed as described previously[51,52] using two compatible plasmids: pUT18c expressing T18 fusions with either ClpX$_{wt}$ or ClpX$_{I265E}$ and pKT25 expressing the T25 fusion with either the Φ11 CI$_{wt}$ or its NTD-CI$_{G131*}$. Both plasmids were co-transformed into *E. coli* BTH101 for the Bacterial Adenylate Cyclase Two-Hybrid (BACTH) system and plated on LB supplemented with ampicillin (100 μg ml$^{-1}$), kanamycin (30 μg ml$^{-1}$), X-gal (20 μg ml$^{-1}$) and IPTG (100 μM). After incubation at 30 °C for 48 h (early reaction) to 72 h (late reaction), the protein−protein interaction was detected by a colour change. Blue colonies represent an interaction between the two clones, while white/yellow colonies are negative for any interaction.

## Promoter activity assay

For the β-lactamase assays, overnight cultures were prepared by inoculating a single colony from each strain into a 5 ml TSB supplemented with 10 μg ml$^{-1}$ erythromycin and incubated at 37 °C, 120 rpm for 16–18 h. The cultures were then diluted 1/50 in 13 ml of fresh TSB supplemented with 10 μg ml$^{-1}$ erythromycin and grown at 37 °C and 120 rpm to early exponential phase (OD$_{540}$-0.15−0.2). 200 μl of culture was added directly to 800 μl of potassium phosphate buffer (50 mM, pH 5.9, supplemented with 10 mM sodium azide) and frozen on dry ice. Where SOS induction of the *cro* promoter was assessed, cultures were split in two (6 ml each), induced either with or without MitC (2 μg ml$^{-1}$) and incubated at 30 °C, 80 rpm until sampling as described above. β-lactamase assays, using nitrocefin as substrate, were performed as described[53,54]: 50 μl of the collected sample were mixed with 50 μl of nitrocefin stock solution (192 μM made in 50 mM potassium phosphate buffer, pH 5.9), and immediately reading the absorbance at 490 nm using an ELx808 microplate reader (BioTek) for 30 min. Promoter activity was calculated as Promoter activity = $(dA_{490}/dt(h))/$ $(OD_{540} \times d \times V)$, where OD$_{540}$ is the absorbance of the sample at OD$_{540}$ at collection, d is the dilution factor, and $V$ is the sample volume. Note that only the linear segment of the resulting absorbance readings is considered for activity calculations.

## Preparation of samples and quantitative whole genome sequencing

Overnight cultures were prepared by inoculating a single colony from each strain into a 5 ml TSB and incubated at 37 °C, 120 rpm for 16–18 h. The desired strains were then diluted 1:50 in 13 ml of fresh TSB and grown to exponential phase *(*OD$_{540}$-0.15−0.20) to collect samples before induction. Next, the cultures were treated with MitC (2 μg ml$^{-1}$) and incubated at 30 °C and 80 rpm for 60 min prior to sample collection. One ml samples were collected, and genomic DNA was extracted using the GenElute Bacterial DNA kit (Sigma Aldrich) according to the manufacturer's instructions. The DNA was precipitated by adding 10% (v/v) 3 M sodium acetate (pH 5.2), 2.5 volumes of 100% ethanol and incubation of the mixture for 1 h at −80 °C. The DNA was then pelleted at 16873 × *g* for 30 min at 4 °C and washed once with 1 ml of ice-cold 70% (v/v) ethanol. After centrifugation, the DNA pellets were air-dried for 30 min and re-suspended in 25 μl of TE buffer (10 mM Tris-HCl, 1 mM EDTA, pH 8.0). Quality control of DNA samples was tested using Agilent Bioanalyzer 2100 and whole genome sequencing (WGS) was performed at the University of Glasgow's Polyomics Facility using Illumina TruSeq DNA Nano library prep, obtaining 2 × 75 bp pair end reads with DNA PCR free libraries.

## Preparation of custom reference genomes

Genomic DNA of the RN450 (NCTC8325-4) reference stock in the lab (JP1250) was extracted and sequenced as described above. Next, reads

were assembled to a scaffold of the deposited NCTC8325 (GenBank Accession CP000253 [https://www.ncbi.nlm.nih.gov/nuccore/87201381]) reference genome using the PATRIC Bioinformatics Resource Center[55]. The three prophages of NCTC8325 were deleted and any mutations identified were curated manually. Sequencing reads were then reassembled to the curated genome (GenBank Accession CP097113 [https://www.ncbi.nlm.nih.gov/nuccore/CP097113]) as described above for verification. The sequences of either Φ11 (RefSeq Accession NC_004615 [https://www.ncbi.nlm.nih.gov/nuccore/NC_004615]) or 80α (RefSeq Accession NC_009526 [https://www.ncbi.nlm.nih.gov/nuccore/NC_009526]) were inserted into attachment sites 5 and 7[49], respectively and correct insertion verified by assembly of genome sequencing reads for strain JP18269 (GenBank Accession CP097114 [https://www.ncbi.nlm.nih.gov/nuccore/CP097114]) or JP18270 (GenBank Accession CP097115 [https://www.ncbi.nlm.nih.gov/nuccore/CP097115]) for Φ11 or 80α lysogens, respectively. The curated genomes were next uploaded to the Galaxy web platform, and we used the public server at usegalaxy.org to analyse the data[56]. Genomes were reannotated using Prokka v1.14.6[57,58] (Galaxy Version 1.14.6+galaxy1).

## Analysis of whole-genome sequencing data

The sequencing data were uploaded to the Galaxy web platform, and we used the public server at usegalaxy.org to analyse the data[56]. The read quality of paired reads was assessed using FastQC v0.11.8[59] (Galaxy Version 0.72+galaxy1) followed by adapter trimming using Trimmomatic v0.38[60] (Galaxy Version 0.38.0) and standard setting for paired-end reads and Illumina data. Trimmed reads were then reassessed using FastQC and mapped to custom genomes of RN450 reference genome containing either prophage Φ11 or 80α using the Burrows-Wheeler Alignment Tool v0.7.17.4[61,62] (Galaxy Version 0.7.17.4) with default settings and saved as BAM files. To normalise sequence coverage across experiments, we first filtered the aligned reads mapping to the bacterial chromosome and not belonging to the prophage using the BAMTools v2.4.0 Filter tool[63] (Galaxy Version 2.4.1). The number of mapped reads for each experiment was extracted from the filtered BAM files using the SAMTools stats utility v1.9[64] (Galaxy Version 2.0.2+galaxy2). Average genome coverage was calculated using the following formula: average genome coverage = (number of mapped reads) × (average read length (bp))/(genome length (bp)). Next, we computed the relative coverage over 50 bp sliding windows along the entire chromosome without normalisation for each of the experiments of the unfiltered BAM files using the bamCoverage tool of the deepTools2 package v3.3.2[65] (Galaxy Version 3.3.2.0.0). These coverage files were saved in bedgraph format and further analysed using RStudio v2021.9.1.372 (Ghost Orchid)[66] and R v4.1.2[67]. Samples were normalised by dividing each coverage window by the average genome coverage calculated for each experiment. Final coverage graphs were plotted in RStudio using ggplot2 v3.3.5[68] and genome organisation around the plotted area extracted from the gff3 file produced by the Prokka annotation and graphed using gggplot2 and the gggenes package v0.4.1[69].

## Determination of prophage integration and excision frequency

Reads were mapped onto either the prophage-free, prophage-containing or circularised phage genomes using BWA[61,62] to determine read coverage of the bacterial *att*B, the prophage left attachment site *att*L and the excised phage *att*P sites, respectively. Reads around the relevant attachment site (±800 bp), which had a matching mate read were extracted using the Samtools view[64] command in Galaxy. The reduced dataset was then further filtered in Rstudio (for filtering scripts, see supplementary material) and only reads in which the paired reads mapped across the attachment site were counted. For a pair to be counted as overlapping, either both reads needed to map clearly to the two different sides of the attachment side or, if the reads contained the attachment side, they needed to begin 10 bp before it started to be considered. Reads were normalised by dividing the reads for each *att*

site by the average genome coverage calculated for each experiment. The fraction of integrated prophage was calculated by dividing the reads mapping to *att*L by the sum of the reads mapping to both *att*L and *att*B. The fraction of circularised and excised phage was calculated by dividing the reads mapping to *att*P by the sum of the reads mapping to *att*P and *att*L.

### Statistical analyses
Statistical analysis was performed as indicated in the figure legend. In general, phage titres were $log_{10}$-transformed and analysed by either One-Way ANOVA followed by Tukey's HSD post-test or using a Student's unpaired two-tailed *t* test as appropriate for the relevant comparison. Promoter activity data were analysed on raw activity data by either One-Way ANOVA followed by Tukey's HSD post test and Bonferroni correction or using a Student's unpaired two-tailed *t* test as appropriate for the relevant comparison. All analysis was done using RStudio.

### Reporting summary
Further information on research design is available in the Nature Portfolio Reporting Summary linked to this article.

## Data availability
All underlying data are provided within the manuscript and raw data as well as statistical analyses performed are provided as Source data files. Reference sequences and whole genome sequencing reads can be accessed through Bioproject PRJNA835099. Source data are provided with this paper.

## Code availability
All underlying, relevant code has been added to the Supplementary Software file.

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

## Acknowledgements

This work was supported by grants MR/M003876/1 and MR/S00940X/1 from the Medical Research Council (MRC, UK; https://mrc.ukri.org), BB/N002873/1 and BB/S003835/1 from the Biotechnology and Biological Sciences Research Council (BBSRC, UK; https://bbsrc.ukri.org), and Wellcome Trust 201531/Z/16/Z (https://wellcome.org), to J.R.P. M.A.T. was funded by a PhD studentship provided by Al Baha University-Kingdom of Saudi Arabia (bu.edu.sa). The funders had no role in study design, data collection and analysis, decision to publish, or preparation of the manuscript.

## Author contributions

A.F.H. and J.R.P. conceived the study and M.A.T. conducted the experiments. M.A.T., J.R.P., and A.F.H. analysed the data. A.F.H. wrote the initial manuscript and all authors contributed to its revisions. Funding was acquired by J.R.P.

## Competing interests

The authors declare no competing interests.
