## [Peer Review File · Nature Communications]

The ClpX protease is essential for inactivating the CI master repressor and completing prophage induction in *Staphylococcus aureus*REVIEWER COMMENTS

Reviewer #1 (Remarks to the Author):

In this work, Thabet et al. tackle the lysis/lysogeny decision of Staphylococcus phages 80alpha and phi11. Guided by indications that the *clp* genes are involved in phage replication, they set about showing that (1) only ClpX and ClpP are involved, (2) they are only involved in induction (not lytic replication), (3) one, ClpP, is primarily involved because it plays a role in activating the bacterial SOS response, (4) the other, ClpX, directly interacts with the NTD of the RecA* triggered cleavage product of the CI (and not the full length version), which is otherwise capable of maintaining phage repression and (5) that this ClpX activity is not dependent on a canonical ClpP-ClpX interaction. The work is excellent – a deep dive into one of the best characterized regulatory events, uncovering something truly new – very well-suited to the journal. It was also done thoroughly, with excellent controls, and a number of elegant experiments: I was particularly impressed with the delta Ori work, and the superinfection immunity assays.

As an expert in bacteriophages, temperate phages, and the lysis/lysogeny decision, I feel qualified to assess every aspect of the manuscript. I was aided in this review by a graduate student.

Critical Concerns: I have no critical concerns.

Major Concerns: I have no major concerns that would require further experiments.

Moderate Concerns:

- The manuscript is somewhat lengthy, with some work (Fig 3-4 especially) feeling repetitive (what is one showing the other isn't? Not clear to me the value of both) and not clearly fitting narratively after Fig S1/Fig 1 show that phage replication proceeds in the absence of the *clp* genes. There is also a little room to trim some repetition throughout the results. Similarly, the value of panels 2A and 2B – presenting the same data, re-arranged, doesn't appear to make two different cases and could be streamlined. The written justification for the whole genome sequencing ("to have a much better picture") seems flimsy.
- Throughout the manuscript, the authors avoid discussing effect sizes – the only way to get at this information is through looking at the figures, and vague language is used in the results. Where possible, specify (e.g. "a 5 log decrease in titre" "restored to wt levels" "a reduction to a mere 2 log pfu/ml" etc). E.g. L256 "virtually absence", L309 "Significantly delayed" (how much? How significant) L309 "and reduced" (how much?). L325 "showed increased levels", L327 "did not increase substantially", L379 "significant increase", L383 "high expression levels", L397 "was repressed", L423 "substantial reduction", L430 "almost completely blocked", L435 "very marginal" L438 "almost completely", L456 "some, but not full", L474 "noticeably increased" ... etc.
- There is a narrative thread that isn't quite woven together about spontaneous induction – the higher basal levels in the ClpP mutant, for instance. The term spontaneous induction is never used – and great points are made (e.g. 244-246, L311-313, L541 etc – that are never fully linked). Some counterpoints for instance, L262 – several phages (incl lambda) show some level of induction in RecA mutants. In L327, the implication is that there is no SOS activation or induction of the phage in the ClpP mutant background – presumably that's readily testable by showing spontaneous phage release (no MitC) assays like those shown in Fig 1? Ultimately, this lack of distinction and unifying these disparate points leads to some confusion e.g. L367 "required for induction" is true in the sense of MitC induction, but not spontaneous induction. Finally, L544-547 – this hypothesis doesn't quite land for me, because no difference in RecA expression is seen in Fig 2 and I don't see why the lack of ClpP would increase RecA expression.
- The word "removing" (e.g. L57) or "elimination" is used repeatedly to refer to the action of ClpX on the NTD of CI. This seems to imply some sort of cleavage, when the authors elsewhere imply that this has more to do with an unfolding of the NTD. Regardless, language throughout should be revised to avoid an explicit mechanism (e.g. "ClpX is required to abolish repression by the NTD of CI"). The unfoldase mechanism can be discussed/proposed in the discussion.
- Abstract is somewhat weak, especially for a generalist audience. "can integrate... thanks to expression ... CI" is strictly true but skips a few steps. "SOS induction" Induction of the bacterial SOS pathway? SoS-driven induction of the phage? Not clear at this point to a lay audience that a phage is being induced at all here. "However, it is unknown how the CI NTD is removed, a process

that is essential to allow prophage induction" could be rewritten "this is necessary, but not sufficient to induce the phage as the NTD is still". "The specific" should be "a specific". Consider revamping abstract.

- L404-407 I'm a little confused here. RecA can still be activated in the absence of LexA cleavage – this is what initiates the SOS response in the first place (the LexA cleavage de-repressing recA magnifies the signal). I'm not clear if the literature consistently shows that phage induction requires the de-repression of RecA, or if "ambient" RecA is sufficient – this work would suggest you need SOS activation (not just RecA activation), but I don't know if that is widely known or true.

Minor Concerns:

- Would be nice to have a tie-in to Lambda, as the go-to model. Is there a role for ClpX or ClpP in CI NTD suppression there? Is it known?
- Figure 8 is not referred to in text.
- Absorbance at 540 seems a little unusual.
- I am unfamiliar with the phage infection assay requiring fresh media and equal amounts of phage buffer – this is not typical with the phages I am more familiar with. Could the authors add a line or two specifying why this is done?
- "MC" is added as an abbreviation somewhat out of nowhere. Might be better to use MitC?
- L267-268 – Why only these two genes early in the SOS response, and nothing "deeper" (e.g. SulA)?
- L323. This is known to only happen in the one phage? The pattern certainly looks like in situ replication and not, for instance, imprecise excision
- L332: "prior to prophage excision" – again, not strictly true re: excision, since some replication happens prior to excision (in at least one of your phages!) - .
- Terminology. Superinfection exclusion (L419) blocks DNA entry into a cell – e.g. SieA, SieB, etc. Superinfection Immunity is repressor based. Correct this.
- L507-510: I didn't follow this at all.

Line-by-Line Comments:

- L5: Replicated should be replicate
 - L11: "encoded divergent to" is awkward.
 - L19: specify "bacterial LexA""
 - L34: "important" = a weak claim. Be more specific.
 - L63-67: Specify Temperature/agitation default?
 - L74: Is that "Sartorius". Specify filter material, quite relevant in phage work.
 - L82: Molten, presumably
 - L96-97: First sentence here can be omitted.
 - L265-66 "different behaviours" not clear exactly what is being referred to here
 - L266: "unique phenotype" not clear exactly which phenotype (and why it is unique). Distinct from the other Clp, perhaps?
 - L276: The comment about the NTD is premature at this point – data for this doesn't coe until later.
 - L292: "phage replication" – Not true, given Fig 1 and S1 – reword.
 - L302: Again, premature – as you allude to later, these could be involved in in situ replication, or excision, or several processes downstream of CI repression.
 - L360: Specify which results (not the Fig 4 results just described)
 - L444: Requires a reference
 - L446: Chapron should be Chaperone
 - L461: omit "on"
 - L472: Extra parentheses/orpha parathesis
 - L474: "its" unclear antecedent.
 - L536: ClpP doesn't inhibit full phage induction, its absence does. Clarify.
- References: 41 and 42 are duplicates. Some references alternate between title case and sentence case.
- L829: Induction is not what is being tested here – replication is.

Reviewer #2 (Remarks to the Author):

The authors describe an important function of the ClpX protease in the inactivation of the CI regulator, required for prophage induction of phages (phi11 and 80alpha) in *Staphylococcus aureus*.

The overall topic is interesting and provides a missing link in our understanding of prophage induction governed by CI-type regulators. They provide compelling data that ClpP and ClpX are involved in prophage induction, but that they inherit different roles. While ClpX was shown to be essential for phage replication and directly interacts with the NTD of CI, ClpP appears to act upstream through the induction of the SOS response. The importance of ClpP for the induction of the cellular SOS response via the degradation of the LexA NTD is well in agreement with previous studies. It is interesting, that ClpX appears to be specific for CI.

The overall manuscript is well written and clearly structured. I have a number of comments and suggestions that I feel need to be addressed:

- 1) While this is a study of broad interest, it nevertheless deserves the host species and phages to be mentioned in the abstract. I find this helpful and important for readers.
- 2) Material and methods: please include the protocol for "lysogenization" of the host strains.
- 3) L.277-279: "...there is always some basal induction of resident prophages, an effect that is not seen in the clpX mutant". I am not convinced that there is absolutely NO spontaneous induction anymore. I still see slight variations in the triplicates, indicating that the values are not "0". Even low frequencies can be physiologically relevant and it would be important to mention.
- 4) Figure 1 and S1, also following: PFU was assessed on lawns; please mention the time of sampling after induction/infection. This is relevant information. I would also suggest to add an "uninduced" control to estimate the rate of spontaneous induction.
- 5) Figure 4: the authors assess the impact of ClpP and ClpX on phage replication and excision. A straightforward and nicely complementary experiment would be the measurement of phage DNA circularization via qPCR as a direct determination of excision.
- 6) Figure 8: The authors suggest a nice model, which is overall supported by their data and I am quite convinced that they are on the right track. However, what I am still missing here in the data set would be clear evidence of the effect of ClpX on CI NTD at the protein level. E.g. by in vitro protein folding studies of CI and Western blot analysis to exclude degradation.

Minor

- L.399: "possible"

- L. 440: "...was no longer cleaved by RecA*" – since this is autocatalytic, this sentence should be revised

Response to reviewers

We would like to thank all reviewers for their positive comments and feedback on our manuscript. We have addressed their concerns point by point below in blue.

Reviewer #1 (Remarks to the Author):

In this work, Thabet et al. tackle the lysis/lysogeny decision of Staphylococcus phages 80alpha and phi11. Guided by indications that the *clp* genes are involved in phage replication, they set about showing that (1) only ClpX and ClpP are involved, (2) they are only involved in induction (not lytic replication), (3) one, ClpP, is primarily involved because it plays a role in activating the bacterial SOS response, (4) the other, ClpX, directly interacts with the NTD of the RecA* triggered cleavage product of the CI (and not the full length version), which is otherwise capable of maintaining phage repression and (5) that this ClpX activity is not dependent on a canonical ClpP-ClpX interaction. The work is excellent – a deep dive into one of the best characterized regulatory events, uncovering something truly new – very well-suited to the journal. It was also done thoroughly, with excellent controls, and a number of elegant experiments: I was particularly impressed with the delta Ori work, and the superinfection immunity assays.

As an expert in bacteriophages, temperate phages, and the lysis/lysogeny decision, I feel qualified to assess every aspect of the manuscript. I was aided in this review by a graduate student.

Critical Concerns: I have no critical concerns.

Major Concerns: I have no major concerns that would require further experiments.

Moderate Concerns:

- The manuscript is somewhat lengthy, with some work (Fig 3-4 especially) feeling repetitive (what is one showing the other isn't? Not clear to me the value of both) and not clearly fitting narratively after Fig S1/ Fig 1 show that phage replication proceeds in the absence of the *clp* genes. There is also a little room to trim some repetition throughout the results. Similarly, the value of panels 2A and 2B

We thank the reviewer for highlighting this and have endeavoured to shorten and remove repetition throughout where possible. Figure 3 has been moved into the supplementary material as it reiterates some data also presented in Figure 4. However, we believe the figure to still add some additional information as it shows a longer time course of induction as well as complementation of the observed defects in the *clpP* and *clpX* mutants. Similarly, we have moved Figure 2A into the supplementary material to reference absolute expression values for the reporters in the different backgrounds. Figure 2B is retained with the main message of the experiment.

– presenting the same data, re-arranged, doesn't appear to make two different cases and could be streamlined. The written justification for the whole genome sequencing (“to have a much better picture”) seems flimsy.

See response above. The rationale for the whole genome experiment has been improved following rearrangement of the data.

- Throughout the manuscript, the authors avoid discussing effect sizes – the only way to get at this information is through looking at the figures, and vague language is used in the results. Where possible, specify (e.g. “a 5 log decrease in titre” “restored to wt levels” “a reduction to a mere 2 log pfu/ml” etc). E.g. L256 “virtually absence”, L309 “Significantly delayed” (how much? How significant) L309 “and reduced” (how much?). L325 “showed increased levels”, L327 “did not increase

substantially”, L379 “significant increase”, L383 “high expression levels”, L397 “was repressed”, L423 “substantial reduction”, L430 “almost completely blocked”, L435 “very marginal”, L438 “almost completely”, L456 “some, but not full”, L474 “noticeably increased” ... etc.

We thank the reviewer for pointing this out and have now included effect sizes in our description of experimental results throughout the manuscript.

- There is a narrative thread that isn't quite woven together about spontaneous induction – the higher basal levels in the ClpP mutant, for instance. The term spontaneous induction is never used – and great points are made (e.g. 244-246, L311-313, L541 etc – that are never fully linked). Some counterpoints for instance, L262 – several phages (incl lambda) show some level of induction in RecA mutants. In L327, the implication is that there is no SOS activation or induction of the phage in the ClpP mutant background – presumably that's readily testable by showing spontaneous phage release (no MitC) assays like those shown in Fig 1? Ultimately, this lack of distinction and unifying these disparate points leads to some confusion e.g. L367 “required for induction” is true in the sense of MitC induction, but not spontaneous induction. Finally, L544-547 – this hypothesis doesn't quite land for me, because no difference in RecA expression is seen in Fig 2 and I don't see why the lack of ClpP would increase RecA expression.

We would like to thank the reviewer for this comment. Indeed, we have now performed the requested experiments and assessed the clpP and clpX mutants' impact on spontaneous prophage induction. We show that both mutants had divergent impacts on spontaneous prophage release with the clpP mutant promoting and the clpX mutant preventing spontaneous prophage release (Fig. 7). The degree of phage release promotion was dependent on the assessed prophage and could be related to overall repressor affinities to the prophage promoter or other genes involved in the individual prophage's lifecycle.

- The word “removing” (e.g. L57) or “elimination” is used repeatedly to refer to the action of ClpX on the NTD of CI. This seems to imply some sort of cleavage, when the authors elsewhere imply that this has more to do with an unfolding of the NTD. Regardless, language throughout should be revised to avoid an explicit mechanism (e.g. “ClpX is required to abolish repression by the NTD of CI”). The unfoldase mechanism can be discussed/proposed in the discussion.

We appreciate this comment and have modified the document accordingly to avoid stating an explicit mechanism.

- Abstract is somewhat weak, especially for a generalist audience. “can integrate... thanks to expression ... CI” is strictly true but skips a few steps. “SOS induction” Induction of the bacterial SOS pathway? SoS-driven induction of the phage? Not clear at this point to a lay audience that a phage is being induced at all here. “However, it is unknown how the CI NTD is removed, a process that is essential to allow prophage induction” could be rewritten “this is necessary, but not sufficient to induce the phage as the NTD is still”. “The specific” should be “a specific”. Consider revamping abstract.

We agree with the reviewer's comment and have modified the abstract to make it more accessible to a generalist audience. Unfortunately, due the word limits imposed by the journal guidelines, we are restricted in the amount that can be placed here. While we appreciate that more detail and background might be beneficial to the abstract, these details are expanded on in the introduction.

- L404-407 I'm a little confused here. RecA can still be activated in the absence of LexA cleavage – this is what initiates the SOS response in the first place (the LexA cleavage de-repressing recA magnifies the signal). I'm not clear if the literature consistently shows that phage induction requires the de-repression of RecA, or if 'ambient' RecA is sufficient – this work would suggest you need SOS activation (not just RecA activation), but I don't know if that is widely known or true.

We apologise for not stating this more accurately. The reviewer is correct in their statement and indeed we believe that this is what enables some phage reproduction in the clpP mutants. We have clarified this throughout the manuscript and expanded the discussion.

Minor Concerns:

- Would be nice to have a tie-in to Lambda, as the go-to model. Is there a role for ClpX or ClpP in CI NTD suppression there? Is it known?

While there is no role for ClpXP in the degradation of the lambda CI protein, several key regulatory proteins involved in the lysis/lysogeny decision process in this model phage are subject to rapid proteolytic degradation. These are therefore rapidly lost once they are no longer synthesised. These instable proteins are O, N, CII, CIII and Xis and are degraded by ClpXP (O-protein), Lon (N-protein and Xis), and FtsH for CII and CIII as reviewed by Casjens SR, Hendrix RW. Bacteriophage lambda: Early pioneer and still relevant. *Virology* 479-480, 310-330 (2015).

- Figure 8 is not referred to in text.

Thank you for highlighting this. Figure 8 has now been cited in the text.

- Absorbance at 540 seems a little unusual.

Historically, absorbance in the protocols of the lab has been measured at 540nm. This was rather for consistency with old data and procedures than making an effective difference to the experiments as under the experimental conditions absorbances at 540 and 600 nm were almost identical. We have since moved to a more standard absorbance of 600 nm.

- I am unfamiliar with the phage infection assay requiring fresh media and equal amounts of phage buffer – this is not typical with the phages I am more familiar with. Could the authors add a line or two specifying why this is done?

Again, this was followed more to be consistent with historical experiments and is not a strict requirement as long as CaCl₂, required for phage adsorption, is supplemented. By resuspending in a mix of fresh medium and phage buffer, we provide sufficient CaCl₂ for successful phage adsorption and infection. The experiment can also be performed by directly supplementing 5 mM CaCl₂ to the culture.

- "MC" is added as an abbreviation somewhat out of nowhere. Might be better to use MitC?

Corrected as suggested.

- L267-268 – Why only these two genes early in the SOS response, and nothing "deeper" (e.g. Sula?)

Our intention was to monitor general activation of the SOS response and we selected two genes early on with which we already had previous experience and that would allow us to assess the impact of ClpP and ClpX on SOS response induction.

- L323. This is known to only happen in the one phage? The pattern certainly looks like *in situ* replication and not, for instance, imprecise excision

In situ replication is known to occur in both prophages tested. However, the degree of *in situ* replication can vary considerable between different prophages and depends on each prophage's induction and excision kinetics with $\Phi 11$ showing pronounced *in situ* replication after 60 min (conditions of our experiments here), while 80α only shows this after 120 min. See also main article and supplementary materials of Chen J, Quiles-Puchalt N, Chiang YN, Bacigalupe R, Filloi-Salom A, Chee MSJ, Fitzgerald JR, Penades JR. 2018. Genome hypermobility by lateral transduction. *Science* 362:207-212.

- L332: "prior to prophage excision" – again, not strictly true re: excision, since some replication happens prior to excision (in at least one of your phages!) - .

Thank you for highlighting this. We have corrected this phrasing and incorporated a more detailed explanation of the staphylococcal prophage induction cycle into the manuscript. "Furthermore, staphylococcal prophages such as $\Phi 11$ or 80α , do not follow the classical excision, replication, and packaging (ERP) cycle, where the prophage first excises from the bacterial genome. Instead, they follow the replication, packaging, and excision (RPE) cycle, where the prophage initiates replication and packaging while still integrated into the bacterial genome and excises late in the induction cycle²⁹. Hence, it was clear that ClpP and ClpX should control the heart of the lysogenic switch system and should control prophage derepression."

- Terminology. Superinfection exclusion (L419) blocks DNA entry into a cell – e.g. SieA, SieB, etc. Superinfection Immunity is repressor based. Correct this.

Thank you for indicating this. We have corrected this error.

L507-510: I didn't follow this at all.

We acknowledge that we have not been clear enough in our description and that this was highly speculative. We have therefore removed the statement and expanded the section to clarify the remaining points.

Line-by-Line Comments:

L5: Replicated should be replicate Corrected

L11: "encoded divergent to" is awkward. Replaced with "encoded in opposite orientation to"

L19: specify "bacterial LexA" Changed

L34: "important" = a weak claim. Be more specific. Replaced with "required"

L63-67: Specify Temperature/agitation default? Added

L74: Is that "Sartorius". Specify filter material, quite relevant in phage work. Changed and additional information added.

L82: Molten, presumably Added

L96-97: First sentence here can be omitted. Removed

L265-66 "different behaviours" not clear exactly what is being referred to here We have clarified this stamen now as follows "This was unlikely, however, since the phenotypes of both mutants were

different (complete inability or reduction of phage progeny production for the *clpX* and *clpP* mutants, respectively) and inconsistent with the known roles of ClpXP in SOS response induction{Cohn, 2011 #2659}, where the concerted action of ClpXP is required for LexA NTD degradation.”

L266: “unique phenotype” not clear exactly which phenotype (and why it is unique). Distinct from the other Clp, perhaps?

Sentence changed, see previous point.

L276: The comment about the NTD is premature at this point – data for this doesn’t come until later.

This section refers to the bacterial LexA protein and its NTD. We believe that in the context of published literature for this protein, it is appropriate to refer to the LexA NTD at this point.

L292: “phage replication” – Not true, given Fig 1 and S1 – reword.

Thank you for pointing this out. We have changed the section title to: “ClpX is essential for prophage derepression.”

L302: Again, premature – as you allude to later, these could be involved in in situ replication, or excision, or several processes downstream of CI repression.

Thank you for this comment. We have reformulated this section to keep the logic of the hypothesis better focused.

L360: Specify which results (not the Fig 4 results just described)

We have referred to Figs. 2 & 3 to clarify this.

L444: Requires a reference

Reference added (Jelsbak L, Ingmer H, Valihrach L, Cohn MT, Christiansen MH, Kallipolitis BH, Frees D. 2010. The chaperone ClpX stimulates expression of *Staphylococcus aureus* protein A by Rot dependent and independent pathways. PLoS One 5:e12752.

L446: Chapron should be Chaperone

Corrected

L461: omit “on”

Done

L472: Extra parentheses/orpha parathesis

Corrected

L474: “its” unclear antecedent.

Changed to “While ClpX did not substantially interact with the full-length CI_{wt}, it interacted more strongly with the CI_{G131*} construct expressing only the CI NTD (Fig. 6c)”

L536: ClpP doesn’t inhibit full phage induction, its absence does. Clarify.

Changed to: “ ClpP on the other hand was shown to be essential for the activation of the bacterial SOS response, and its absence consequently prevented full phage induction.”

References: 41 and 42 are duplicates. Some references alternate between title case and sentence case.

These are two different references by the same group referring to short and long-read alignment, respectively of the BWA method. We have left case formatting the same as was used in the published articles.

L829: Induction is not what is being tested here – replication is.
Changed to “ClpP and ClpX impact phage replication”

Reviewer #2 (Remarks to the Author):

The authors describe an important function of the ClpX protease in the inactivation of the CI regulator, required for prophage induction of phages (phi11 and 80alpha) in *Staphylococcus aureus*.

The overall topic is interesting and provides a missing link in our understanding of prophage induction governed by CI-type regulators. They provide compelling data that ClpP and ClpX are involved in prophage induction, but that they inherit different roles. While ClpX was shown to be essential for phage replication and directly interacts with the NTD of CI, ClpP appears to act upstream through the induction of the SOS response. The importance of ClpP for the induction of the cellular SOS response via the degradation of the LexA NTD is well in agreement with previous studies. It is interesting, that ClpX appears to be specific for CI.

The overall manuscript is well written and clearly structured. I have a number of comments and suggestions that I feel need to be addressed:

1) While this is a study of broad interest, it nevertheless deserves the host species and phages to be mentioned in the abstract. I find this helpful and important for readers.

We thank the reviewer for highlighting this and have now incorporated these details into the abstract.

2) Material and methods: please include the protocol for “lysogenization” of the host strains.

Thank you for requesting this additional information. A paragraph has been added to the Materials and Methods section of the manuscript.

3) L.277-279: “...there is always some basal induction of resident prophages, an effect that is not seen in the clpX mutant”. I am not convinced that there is absolutely NO spontaneous induction anymore. I still see slight variations in the triplicates, indicating that the values are not "0". Even low frequencies can be physiologically relevant and it would be important to mention.

Thank you for highlighting this. Please see our response to reviewer one regarding this question.

4) Figure 1 and S1, also following: PFU was assessed on lawns; please mention the time of sampling after induction/infection. This is relevant information. I would also suggest to add an “uninduced” control to estimate the rate of spontaneous induction.

We have expanded the materials and methods section to provide the requested information. Generally, we did not perform time course experiments to evaluate phage titres unless specified in the figures and all data reflect phage titres in lysates after overnight incubation on the bench. The rate of spontaneous induction has now been estimated in an additional set of experiments and the results and discussion section modified accordingly.

5) Figure 4: the authors assess the impact of ClpP and ClpX on phage replication and excision. A straightforward and nicely complementary experiment would be the measurement of phage DNA circularization via qPCR as a direct determination of excision.

We thank the reviewer for suggesting this experiment and agree that this would provide additional, complimentary data. We have used the existing whole genome sequencing data to obtain the requested information bioinformatically by mapping reads across chromosomal *attB*, *attL* and circular phage *attP* sites to quantify the level of bacterial chromosomes that had undergone full prophage excision and the amount of circular phage present in the sample (Table 1).

6) Figure 8: The authors suggest a nice model, which is overall supported by their data and I am quite convinced that they are on the right track. However, What I am still missing here in the data set would be clear evidence of the effect of ClpX on CI NTD at the protein level. E.g. by in vitro protein folding studies of CI and Western blot analysis to exclude degradation.

We appreciate the reviewer's valuable comment and agree that additional protein studies would strengthen our findings. In response to this suggestion, we performed western blot analysis to investigate the effect of ClpX on CI NTD at the protein level. Specifically, we generated plasmids expressing N-terminally 3X-FLAG-tagged versions of Phi11 CI_{wt} , CI_{G131E} , and CI_{G131*} and introduced them into both the wild-type (wt) and $\Delta clpP$ or $\Delta clpX$ backgrounds. We subsequently performed western blotting experiments.

Although we successfully detected both CI_{wt} and CI_{G131E} proteins, we did not observe the presence of the tagged post-cleavage fragment. Furthermore, when we assessed the ability of these tagged proteins to block phage infection, as shown in Figure 5 of the revised manuscript, we noticed that the tagged repressors, unlike the untagged repressors, were unable to fully block phage infection. We made attempts to address this issue by altering the tag (using 3X-FLAG, HA-, GFP, and mCherry) and varying the expression levels (evaluating two inducible promoters and one constitutive promoter). However, none of these modifications allowed us to recover the phenotypes observed with the untagged versions of the repressor. Notably, constructs utilizing the 80 α repressor exhibited similar phenotypes to those of $\Phi 11$.

From these results, we propose that the N-terminus of the DNA-binding region of the CI repressor likely contributes to the stability of the CI-NTD after autocleavage and its modification with an N-terminal tag results in its rapid turnover. While we acknowledge that we were unable to fully address this aspect, we believe that the data presented in our study still provide support for our overall model. We have discussed the limitations of our study in the manuscript.

We sincerely appreciate the reviewer's input, as it has highlighted an area for further investigation in our future studies.

Minor

- L.399: "possible"

Changed as suggested.

- L. 440: "...was no longer cleaved by RecA*" – since this is autocatalytic, this sentence should be revised

Changed to "...confirming that autocatalytic cleavage of CI could no longer be triggered by RecA*"

REVIEWERS' COMMENTS

Reviewer #2 (Remarks to the Author):

The revised manuscript has effectively incorporated feedback from both reviewers, addressing their comments appropriately. Notably, the authors have evaluated the effects of clpP and clpX mutants on spontaneous prophage induction, and these findings have been added in the manuscript.

They did not succeed in analysing the impact of ClpX on CI NTD at the protein level. However, explanations in their response letter appear reasonable by pointing out that the N-terminus of the DNA-binding region of the CI repressor could likely contribute to the stability of the CI-NTD after autocleavage and its modification with an N-terminal tag results in its rapid turnover.

Overall, the manuscript has been substantially improved, incorporating valuable insights and discussing limitations and open questions.

Reviewer #3 (Remarks to the Author):

The work done by Thabet et al. describes an important involvement of clpX protease in *Staphylococcus aureus* prophage induction. The findings are significant and will contribute greatly to understanding the fundamental biology involved in the lytic/lysogenic decision-making process. They show that ClpX and ClpP are involved in prophage induction but via different interactions. ClpP plays an important role in activating the bacterial SOS response and ClpX is essential for phage replication and interacts directly with the NTD of the RecA* cleaved product of CI. Methodologies used are elegant and excellently controlled for.

Overall, the manuscript is well written, and I appreciate the slight restructuring of the figures to make the narrative clearer.

Authors have addressed all of our previous comments. I have no further major/moderate comments. I recommend accepting the revised manuscript. Below are a few minor comments/edits.

Minor edits/comments

- Would be interesting to see an elaboration on mutants' involvement in spontaneous induction. For example, line 416-417: is reduction in spontaneous induction a stable phenotype? Was this phenotype tested in more than one lysogen or was just one lysogen created for each mutant? Perhaps a little elaboration (in addition to line 517-519) in the gene's involvement in spontaneous induction in the discussion would be beneficial. Perhaps tie it back to the statement made earlier on line 486-489.

Inconsistencies with terminology

- It may be easier to define spontaneous induction rather than interchanging between "spontaneous induction" and "prophage induction without MitC".

Inconsistencies with effect sizes:

- Line 121: instead of referring to the number of clpX plaque production, refer it to as reduction in phage titre compared to wildtype as was done with clpP
- Line 254: define "appreciable drop"

Methodology:

- Curious as to why different O.Ds within the exponential phase were used. For instance, O.D 1.5 for the phage infection assay and O.D. 0.35 for lysogenisation of phages.

Line by line comments

- 104: "or" instead of "and"
- 107: removal of "in the"
- Line 193: "is" instead of "as"
- Line 235 "notable" instead of "notably"
- Line 263: phage kinetics is too vague

Response to reviewers

We would like to thank all reviewers for the time taken to evaluate our manuscript and their positive comments and feedback. We have addressed their concerns point by point below in blue.

Reviewer #2 (Remarks to the Author):

The revised manuscript has effectively incorporated feedback from both reviewers, addressing their comments appropriately. Notably, the authors have evaluated the effects of clpP and clpX mutants on spontaneous prophage induction, and these findings have been added in the manuscript. They did not succeed in analysing the impact of ClpX on CI NTD at the protein level. However, explanations in their response letter appear reasonable by pointing out that the N-terminus of the DNA-binding region of the CI repressor could likely contribute to the stability of the CI-NTD after autocleavage and its modification with an N-terminal tag results in its rapid turnover.

Overall, the manuscript has been substantially improved, incorporating valuable insights and discussing limitations and open questions.

We thank the reviewer for their positive feedback.

Reviewer #3 (Remarks to the Author):

The work done by Thabet et al. describes an important involvement of clpX protease in *Staphylococcus aureus* prophage induction. The findings are significant and will contribute greatly to understanding the fundamental biology involved in the lytic/lysogenic decision-making process. They show that ClpX and ClpP are involved in prophage induction but via different interactions. ClpP plays an important role in activating the bacterial SOS response and ClpX is essential for phage replication and interacts directly with the NTD of the RecA* cleaved product of CI. Methodologies used are elegant and excellently controlled for. Overall, the manuscript is well written, and I appreciate the slight restructuring of the figures to make the narrative clearer.

Authors have addressed all of our previous comments. I have no further major/moderate comments. I recommend accepting the revised manuscript. Below are a few minor comments/edits.

We thank the reviewer for their positive feedback.

Minor edits/comments

- Would be interesting to see an elaboration on mutants' involvement in spontaneous induction. For example, line 416-417: is reduction in spontaneous induction a stable phenotype? Was this phenotype tested in more than one lysogen or was just one lysogen created for each mutant? Perhaps a little elaboration (in addition to line 517-519) in the gene's involvement in spontaneous induction in the discussion would be beneficial. Perhaps tie it back to the statement made earlier on line 486-489.

Thank you for this interesting comment. While we have performed these experiments in a single lysogen, the data are consistent with data observed from multiple lysogens created after generating

the *clpX* and *clpP* mutants in different strain backgrounds. We therefore are convinced that this is a stable phenotype as the reduction in titres of the *clpX* mutant is consistent at the same low level among these different strain backgrounds and isolates.

We have expanded the final statement of the discussion to tie back to the earlier statement.

“In addition to their role in SOS-mediated prophage induction, both ClpP and ClpX are also important for the levels of spontaneous prophage release. ClpX likely acts in a similar capacity in both SOS and spontaneous prophage induction as phage titres remained comparable either with or without MitC. By contrast, ClpP appears to perform additional regulatory functions controlling stable prophage integration in the bacterial chromosome or preventing erroneous induction events through regulating the levels of additional proteins involved in the lysogenic switch decision.”

Inconsistencies with terminology

- It may be easier to define spontaneous induction rather than interchanging between “spontaneous induction” and “prophage induction without MitC”.

Thank you for the comment. We appreciate that this might be the case in some scenarios. However, we felt that here, we assess different mechanisms where absence of MitC serves as a control for no induction (i.e. for reporter plasmids). The only instance where we refer to spontaneous induction is in the relevant sections and in this instance to define it as phage release without the addition of MitC. We therefore believe that to maintain clarity for the other experiments, it is beneficial to retain the current usage and formulation.

Inconsistencies with effect sizes:

- Line 121: instead of referring to the number of *clpX* plaque production, refer it to as reduction in phage titre compared to wildtype as was done with *clpP*

Changed as requested.

- Line 254: define “appreciable drop”

Thank you for indicating this. We have now calculated the average coverage ratio across the phage genome between induced and uninduced samples and clarified the section.

“In the wt RN450 strain background lysogenic for the Φ 11 and 80 α ori mutants (JP20045 and JP20046, respectively), we observed an appreciable drop in read coverage in the prophage region of the induced ori mutants in both Φ 11 (2.6-fold, ratio induced to uninduced of 0.385) and 80 α (1.6-fold, ratio induced to uninduced of 0.641) (Fig. 3), indicating their inability to replicate once excised from the chromosome.”

Methodology:

- Curious as to why different O.Ds within the exponential phase were used. For instance, O.D 1.5 for the phage infection assay and O.D. 0.35 for lysogenisation of phages.

Thank you for noticing this. The different OD used for lysogenisation is primarily to ensure the formation of a good and confluent lawn of bacteria allowing selection of lysogens. When we do phage infection and induction assays, we have noticed that these are more efficient when using a slightly

lower OD for some phages. This is potentially linked to phage induction kinetics but might also point to some growth phase dependency of the induction cascade, which we are interested in clarifying in the future.

Line by line comments

- 104: "or" instead of "and"

Changed as requested.

- 107: removal of "in the"

Changed as requested.

- Line 193: "is" instead of "as"

Changed to "was" to remain consistent with the rest of the paragraph.

- Line 235 "notable" instead of "notably"

Changed as requested.

- Line 263: phage kinetics is too vague

Changed to "and might reflect differences in the speed of 80 α derepression and/or initial replication compared to Φ 11".